

# The Iceland-Faroe warm-water flow towards the Arctic estimated from satellite altimetry and in situ observations

Bogi Hansen[1], Karin Margretha Húsgarð Larsen[1], Hjálmar Hátún[1], Steffen Malskær Olsen[2], Andrea Martina Ulrike Gierisch[2], Svein Østerhus[3], Sólveig Rósa Ólafsdóttir[4]

[1]Faroe Marine Research Institute, Tórshavn, Faroe Islands.
[2]Danish Meteorological Institute, Copenhagen, Denmark
[3] NORCE Norwegian Research Centre and Bjerknes Centre for Climate Research, Bergen, Norway
[4]Marine and Freshwater Research Institute, Hafnarfjörður, Iceland.

*Correspondence to*: Bogi Hansen (bogihan@hav.fo)

**Abstract.** The inflow of warm and saline Atlantic water to the Arctic Mediterranean (Nordic Seas and Arctic Ocean) between Iceland and the Faroes (IF-inflow) is the strongest Atlantic inflow branch, in terms of volume transport, and associated with a large transport of heat towards the Arctic. The IF-inflow is monitored on a section east of the Iceland-Faroe Ridge (IFR) by use of Sea Level Anomaly (SLA) data from satellite altimetry, a method that has been calibrated by in situ observations gathered over two decades. Monthly averaged surface velocity anomalies calculated from SLA data were

strongly correlated with anomalies measured by moored Acoustic Doppler Current Profilers (ADCPs) with consistently higher correlations when using the reprocessed SLA data released in December 2021 rather than the earlier version. In contrast to the earlier version, the reprocessed data also had the correct conversion factor required by geostrophy. Our results show that the IF-inflow crosses the IFR in two separate branches. The Icelandic branch is a jet over the Icelandic slope with average surface speed exceeding 20 cm s$^{-1}$, but it is narrow and shallow with an average volume transport less than 1 Sv ($10^6$

m$^3$ s$^{-1}$). Most of the Atlantic water crosses the IFR close to its southernmost end in the Faroese branch. Between these two branches, water from the Icelandic branch turns back onto the ridge in a retroflection with a recirculation over the northernmost bank on the IFR. Combining multi-sensor in situ observations with satellite SLA data, monthly mean volume transport of the IF-inflow has been determined from January 1993 to December 2021. The IF-inflow is part of the Atlantic Meridional Overturning Circulation (AMOC), which is expected to weaken under continued global warming. Our results

show no weakening of the IF-inflow. Annually averaged volume transport of Atlantic water through the monitoring section had a statistically significant (95 % confidence level) increasing trend of (0.12±0.10) Sv per decade. Combined with increasing temperature, this caused an increase of 13 % in the heat transport, relative to 0 °C, towards the Arctic of the IF-inflow over the 29 years of monitoring. The near-bottom layer over most of the IFR is dominated by cold water of Arctic origin that may contribute to the overflow across the ridge. Our observations confirm a dynamic link between the overflow

and the Atlantic water flow above. The results also provide support for a previously posed hypothesis that this link may explain the difficulties in reproducing observed transport variations of the IF-inflow in numerical ocean models, with consequences for its predictability under climate change.



# 1 Introduction

Between Iceland and the Faroes (Faroe Islands), there is a flow of relatively warm and saline water in the near-surface layer
from the Iceland Basin into the Norwegian Basin, across the Iceland-Faroe Ridge, "*IFR*", which is part of the Greenland-
Scotland Ridge (Fig. 1). Following tradition, the areas southwest of the ridge are referred to as the "*Atlantic*", whereas the
areas northeast of the ridge are referred to as the "*Arctic Mediterranean*" (Nordic Seas and Arctic Ocean). The warm water
flowing over the IFR is referred to as "*Atlantic water*" and the flow as a whole as the "Iceland-Faroe Atlantic water inflow to
the Nordic Seas" or just "*IF-inflow*". After crossing the IFR, the IF-inflow continues into the Norwegian Basin where it
meets colder and less saline water from various parts of the Arctic Mediterranean, which we collectively refer to as "*Arctic
water*". The boundary between the Atlantic and the Arctic waters is the "*Iceland-Faroe Front*" (Fig. 1), which in the surface
is located northeast of the IFR (Hansen and Meincke, 1979), but slopes so that it hits the top of the ridge (e.g., Tait et al.,
1967; Meincke, 1978). This means that the bottom layer over most the IFR typically is covered by Arctic water and some of
this Arctic water crosses the IFR to pass into the Iceland Basin as "*IFR-overflow*" (Knudsen, 1898).

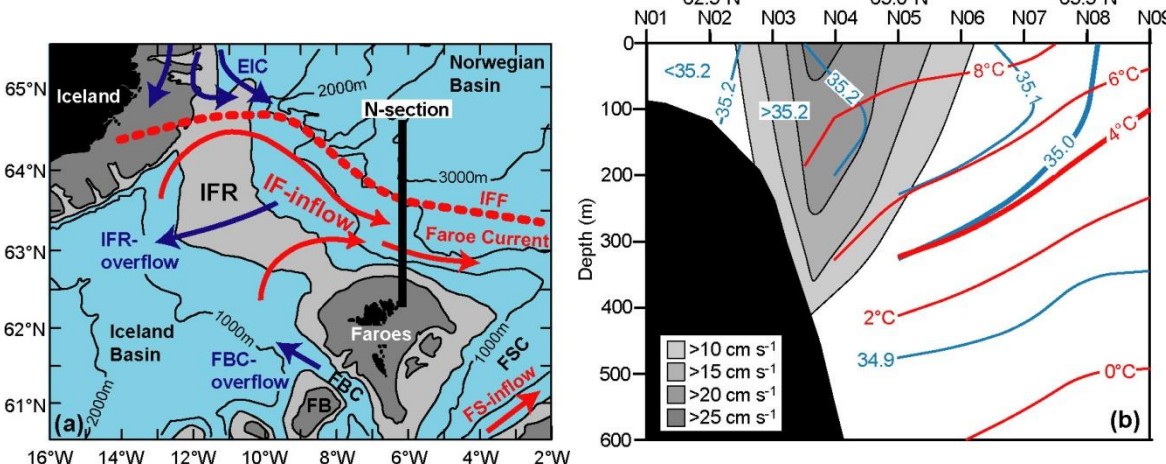

**Figure 1. (a)** The region between Iceland and the Scottish shelf with the main current systems. Dark grey areas are shallower than 200 m,
light grey areas are shallower than 500m. The red arrows indicate the two main branches of warm Atlantic water inflow to the Arctic
Mediterranean. The IF-inflow crosses the Iceland-Faroe Ridge (IFR), meets colder waters of Arctic origin in the Iceland-Faroe Front (IFF),
gets focused into the Faroe Current, and passes through the N-section (black line), where it is monitored. The other main inflow branch,
the FS-inflow, passes through the Faroe-Shetland Channel (FSC) and over the shelf areas west of Scotland. Dark blue arrows indicate
flows of cold water of Arctic origin. The East Icelandic Current (EIC) flows southwards in the upper layers east of Iceland and meets the
IF-inflow in the frontal zone. The Faroe Bank Channel overflow (FBC-overflow) flows through the depths of the Faroe-Shetland Channel
and the Faroe Bank Channel to pass into the Iceland Basin. The IFR-overflow crosses the IFR in various locations close to the bottom. **(b)**
Average conditions on the southern part of the N-section (standard station numbers on top, for reference, see Sect. 2). Red and blue lines
show average isotherms and isohalines, respectively, for the 1989 – 2018 period redrawn from Hansen et al. (2020). The grey-shaded areas
illustrate the average eastward velocity based on (non-simultaneous) ADCP data, redrawn from Hansen et al. (2019).

In addition to the IF-inflow, there is an inflow of Atlantic water west of Iceland (Jónsson and Valdimarsson, 2012) and
one between the Faroes and the European continent, most of which passes through the Faroe-Shetland Channel (Berx et al.,
2013) as the "FS-inflow" (Fig. 1). For the period 1993–2015, Østerhus et al. (2019) combined observational evidence to





estimate the average total volume transport of all the Atlantic inflow branches to 8.0 Sv. With an average volume transport of (3.8±0.5) Sv (Hansen et al., 2015), the IF-inflow, thus, accounts for 48 % of the total, on average.

The presence of warm Atlantic water between Iceland and the Faroes has been known for a long time (e.g., Nielsen, 1904) and the three multi-ship surveys during the ICES-Overflow expedition in 1960 (Tait et al., 1967) showed warm and saline water roughly covering the whole region southwest of the dashed line labelled IFF in Fig. 1 in the surface. In their paper on the Norwegian Sea, Helland Hansen and Nansen (1909) also show the IF-inflow clearly and Hermann (1949) estimated its transport as 4.5 Sv. Despite this, many circulation maps during the latter half of the 20th century show most or even all the Atlantic inflow between Iceland and Scotland to pass through the Faroe-Shetland Channel (e.g., Worthington, 1970; McCartney and Talley, 1984). A more balanced overview of the relative strengths of the various inflows emerged after direct current measurements for the various branches allowed more rigorous transport estimates (e.g., Hansen and Østerhus, 2000).

The Atlantic water approaching the IFR from the Iceland Basin is not as warm and saline as the inflow between the Faroes and Europe, and it is further cooled and freshened by its passage across the IFR (Larsen et al., 2012). Nevertheless, the high volume transport of the IF-inflow means that it carries a lot of heat (Tsubouchi et al., 2021) and salt into the Arctic Mediterranean. Systematic monitoring of its transport and properties has therefore long been recognized as an important task. Regular monitoring of the hydrographic properties of the IF-inflow was initiated in the late 1980s along a standard section, the "N-section", which runs along 6.08° W (Fig. 1). Since 1988, Conductivity Temperature Depth (CTD) observations have typically been carried out three to four times a year. After some preliminary test-deployments, three ADCP moorings were deployed along the N-section in June 1997. This initiated a period during which the IF-inflow was monitored by the regular CTD cruises combined with three to five ADCPs deployed at fixed locations along the section continuously except for annual servicing periods of two to three weeks (Hansen et al., 2003).

The choice of using ADCPs rather than single-point current meters (e.g., Aanderaa) on traditional moorings was made because of the heavy fisheries activity in the region. Over the Faroe slope, the ADCPs were deployed on the bottom in trawl-protected frames. In deeper waters, they were deployed below typical trawling depth in the top of traditional moorings. This prevented heavy equipment loss, but the ADCPs do not measure velocity close to the surface and they give no direct information on the hydrographic properties of the water column except from auxiliary sensors at the instrument.

This meant that the temperature and salinity distribution on the section was only known from the three to four CTD cruises each year. This lack made it difficult to calculate heat transport, but also made it difficult to distinguish the time variability of the Atlantic water from Arctic water coverage and fraction on the section. Even before the Atlantic water passes onto the IFR, it meets Arctic water in the IFR-overflow and mixes with it west of the ridge (Meincke, 1972). Regardless of the location of a monitoring section, it will always contain Arctic water as well as Atlantic water. To determine the transport of Atlantic water through the section, the Atlantic water has to be distinguished from the Arctic water. That is most easily done by using the hydrographic properties since the Atlantic water is warmer and more saline than the Arctic water.



Traditional water mass analysis (e.g., Hermann, 1967) may be used to determine the fraction of Atlantic water at a specific location from its temperature and salinity. Combined with the velocity field, this allows calculation of Atlantic water transport (Hansen et al., 2003). This method requires, however, that there are not more than two different Arctic water types that mix with the Atlantic water and that the source water characteristics are well defined. These requirements are usually not fulfilled in the Iceland-Faroe region. The core of the Atlantic water on the N-section usually has a temperature close to 8 °C

and is underlain by Arctic water with temperature close to 0 °C (Fig. 1b). For the purpose of volume transport calculations, it was therefore decided to define the lower boundary of the Atlantic water extent on the section by the 4 °C isotherm, slightly modified for long-term variations (see Appendix A), whereas the 35.0 isohaline was used to define the northern boundary (Hansen et al., 2015).

     The ADCP-based monitoring system was maintained for almost two decades, but it was demanding to maintain and

instrument failure or loss introduced gaps and inaccuracies into the time series. Volume transport, determined from this system, was also found to be correlated with data from satellite altimetry (Hansen et al., 2010). It was therefore decided to switch monitoring strategy from an ADCP-based to an altimetry-based system. The new system was justified and described by Hansen et al. (2015) and has since then been refined in two technical reports (Hansen et al., 2019; Hansen et al., 2020) with the algorithms summarized in Appendix A.

The basic premise for using an altimetry-based system is that the surface velocity in a given direction, horizontally averaged over an interval perpendicular to that direction is proportional to the difference in sea level height between both ends of the interval. For this to be valid, geostrophy must be assumed and the time scale must be sufficiently long. Since there is a large data set from ADCP and other in situ observations on the N-section, they allow us to check this premise. This became especially important after the Copernicus Marine Environment Monitoring Service (CMEMS) in December 2021

released a new gridded data set, where Sea Level Anomaly, "*SLA*", data for the whole altimetry period had been reprocessed. Also, a new version of the Mean Dynamic Topography, "*MDT*", (Mulet et al., 2021) was released. Checking the accuracy of surface velocity derived from gridded altimetry is one of the main objectives of this study, and in Sect. 3 we compare surface velocities derived from in situ observations to those derived from both the "*old*" (pre-December 2021) and the "*new*" altimetry data.

Once validated, data from satellite altimetry also provide irreplaceable information on the whole flow system in the region. In Sect. 4, we combine altimetry data with data from surface drifters and ADCPs to map the surface flow of the Atlantic water from the Iceland Basin all the way through the N-section. This is followed by a more detailed analysis of the Atlantic water flow across the IFR in Sect. 5. Combining altimetry data with measurements from four ADCP deployments on the IFR, we map the average flow pattern and its variations. One motivation for that is to reconcile the conflicting views

of Orvik and Niiler (2002) versus those of Rossby et al. (2009) on where most of the Atlantic water crosses the IFR.

     An additional objective of this work is to provide updated transport time series of the IF-inflow and discuss their accuracy and their implications. The revised monitoring system described by Hansen et al. (2015) generates values for volume transport and heat transport relative to 0 °C for every month since January 1993. The values are generated from SLA



data by algorithms (see Appendix A) that have been developed by comparing altimetry data with in situ observations. With the new, reprocessed, version of SLA data, it became necessary to reanalyse these relationships and update the algorithms. This also provided the opportunity to quality check the monitoring system as reported in Sect. 6.

Studies on the representation of Atlantic inflow in global climate models (e.g., Heuzé and Årthun, 2019), in hindcast ocean models (e.g., Olsen et al., 2016), and even in ocean reanalyses (Mayer et al., accepted) have demonstrated differences between models and observations, especially for the IF-inflow. Olsen et al. (2016) have suggested that a major reason for this is the inability of models to simulate the coupling between IFR-overflow and IF-inflow over the IFR, even in models with relatively high resolution. In this study, we do not address the IFR-overflow per se, but our results provide added information in support of the hypothesis presented by Olsen et al. (2016), as discussed in Sect. 7.6.

After this introductory section, Sect. 2 presents an overview of the data and statistical methods used in this study. That is followed by the four "results sections": Sect. 3 on deriving surface velocity anomalies from altimetry data, Sect. 4 on the large-scale surface circulation, Sect. 5 on the Atlantic water flow across the IFR, and Sect. 6 on the transport monitoring system. The results reported in these four sections are discussed in Sect. 7, which also presents updated transport time series and discusses their implications. Details of algorithms are elaborated in Appendix A.

## 2 Material and methods

### 2.1 Temperature and salinity data

Since the late 1980s, the fourteen standard stations on the N-section, labelled N01 to N14, (Fig. 2) have typically been occupied 3–4 times a year on CTD cruises and for some of the stations more often. This has resulted in many CTD profiles at each of the stations (Table 1).

**Table 1.** The fourteen standard stations on the N-section are located equidistantly every 10 nautical miles from 62.33° N to 64.50° N along the 6.08° W meridian (except for N14, which is at 6.00° W). Below are listed the bottom depths and number of CTD profiles acquired at each station 1987 – 2019.

| Station: | N01 | N02 | N03 | N04 | N05 | N06 | N07 | N08 | N09 | N10 | N11 | N12 | N13 | N14 |
|---|---|---|---|---|---|---|---|---|---|---|---|---|---|---|
| Bottom (m): | 85 | 119 | 189 | 555 | 1700 | 1950 | 1730 | 1823 | 2169 | 2915 | 3427 | 3200 | 3380 | 3300 |
| Number of profiles: | 155 | 152 | 142 | 135 | 133 | 120 | 122 | 117 | 116 | 114 | 110 | 100 | 98 | 100 |

In addition to the CTD data, we use bottom temperature measurements from the ADCP at site NE (Fig. 2b), and data from two PIES (Pressure Inverted Echo Sounders) that were deployed on the bottom at the locations of standard stations N05 for 645 days and at N07 for 594 days in 2017–2019 (Fig. 2b). The PIES data include measurements of bottom pressure every 30 minute and two-way travel time every 2.5 minute, both of which depend on the temperature profile. The data were quality controlled and averaged to give daily estimates of the travel time corrected for sea level variations as described in Hansen et al. (2020).



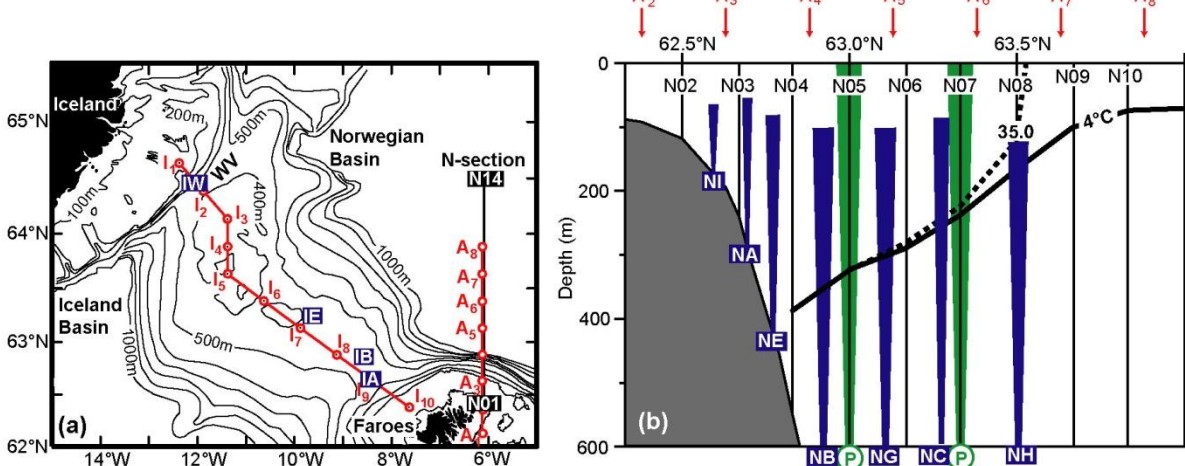

**Figure 2. (a)** Topographical map of the IFR with the northernmost pass, Western Valley, indicated by "WV". Blue rectangles, labelled IA, IB, IE, and IW, show locations of four ADCP moorings. Red circles, labelled $I_1$ to $I_{10}$, show ten altimetry grid points on the IFR connected by a red line roughly following the crest of the ridge. The N-section is shown as a black line with the southernmost, N01, and the northernmost, N14, standard stations indicated. Red circles, labelled $A_1$ to $A_8$, show the altimetry points used for monitoring transport through the N-section. **(b)** The southern part of the N-section with bottom topography in grey. CTD standard stations are indicated by black lines labelled N02 to N10. Locations of seven ADCP sites are marked by blue cones that indicate the typical range. Green cones indicate the locations of two PIES deployments. Altimetry grid points $A_2$ to $A_8$ are marked by red arrows and the thick black lines indicate the average depth of the 4 °C isotherm (continuous) and the 35.0 isohaline (dashed) on the section based on Fig. 1b.

## 2.2 ADCP observations

We use ADCP measurements from seven sites along the N-section and four sites on the IFR (Fig. 2). The ADCP has either been mounted in a buoy at the top of a traditional mooring or within a specially developed frame that protects it from fishing activity. The seven ADCP sites along the N-section are indicated on Fig. 2b with details listed in Table 2. ADCP data from five of these sites were reported in Hansen et al. (2015), but only up to May 2014. At sites NA, NB, and NG, additional data have been acquired and two new sites (NI and NH) have been occupied by one deployment at each site. For the four ADCP sites on the IFR, we only have data from one deployment at each site (Table 2).

The velocity data from the ADCPs are structured into "bins" (depth intervals), which in our case have been either 10 m or 25 m. The ADCPs have been programmed to store data (ensembles) every 20 minutes. The raw data have been quality controlled, de-tided, and averaged to daily values (e.g., Hansen et al., 2017) and the velocity profile linearly interpolated to meter interval.

Due to limited range and side-lobe reflection, an upward-looking ADCP cannot measure the velocity close to the surface and the number of bins with good data for the daily averaged profile varies somewhat from day to day. In this region we find, however, high correlations between the topmost bins (Fig. 3a). This implies that the velocity in a given direction at depth $z$ and time $t$, $u(z,t)$, to a good approximation is proportional to the velocity at a greater depth $z_t$:



$$u(z,t) = \frac{\alpha_0(z)}{\alpha_0(z_t)} \cdot u(z_t, t) \qquad (1)$$


where the "***extrapolation factor***", $\alpha_0(z)$, is a function of depth for each ADCP site, which may be determined by regression analysis. If the ADCP data on a specific day are error-free up to a depth $z_t$, this allows the velocity profile for that day to be extrapolated up to the "***Top depth***" for that site, defined as the uppermost level with good data from the site (Table 2). The procedure is described in more detail in Hansen et al. (2019) and is illustrated by an example in Fig. 3a.


**Table 2.** Main characteristics of the measurements at the eleven ADCP sites with positions, bottom depths, measurement period, number of deployments, number of days, top depth, and location within altimetry interval. At sites NI, NA, NE, IA, IB, and IW the ADCP was in a trawl-protected frame deployed on the bottom. At the other sites, the ADCP was mounted in a buoy on top of a traditional mooring, usually between 600 and 700 m depth, except for site IE, which was protected from fisheries by proximity to a submarine cable.

| Site | Position | | Bottom depth | Measurement period | Number of depl. | Number of days | Top depth | Altimetry interval |
|------|----------|-----------|--------------|--------------------|-----------------|----------------|-----------|--------------------|
| | Latitude | Longitude | | | | | | |
| NI | 62.58° N | 6.08° W | 156 m | Jun 2017 to May 2018 | 1 | 342 | 39 m | $A_2$–$A_3$ |
| NA | 62.70° N | 6.08° W | 300 m | Jun 1996 to May 2015 | 20 | 6663 | 35 m | $A_3$–$A_4$ |
| NE | 62.79° N | 6.08° W | 455 m | Jul 2000 to May 2011 | 8 | 2729 | 73 m | $A_3$–$A_4$ |
| NB | 62.92° N | 6.08° W | 925 m | Oct 1994 to May 2018 | 24 | 7272 | 72 m | $A_4$–$A_5$ |
| NG | 63.10° N | 6.08° W | 1815 m | Jul 2000 to May 2015 | 14 | 4788 | 63 m | $A_4$–$A_5$ |
| NC | 63.27° N | 6.08° W | 1730 m | Oct 1994 to Jun 2000 | 5 | 1517 | 61 m | $A_5$–$A_6$ |
| NH | 63.50° N | 6.08° W | 1802 m | Jun 2015 to May 2016 | 1 | 339 | 65 m | $A_6$–$A_7$ |
| IA | 62.64° N | 8.45° W | 498 m | Sep 2004 to May 2005 | 1 | 259 | 50 m | $I_8$–$I_{10}$ |
| IB | 62.86° N | 8.59° W | 495 m | Jul 2003 to Jun 2004 | 1 | 342 | 45 m | $I_8$–$I_9$ |
| IE | 63.25° N | 9.80° W | 490 m | Jun 2020 to May 2021 | 1 | 343 | 98 m | $I_6$–$I_8$ |
| IW | 64.45° N | 12.06° W | 402 m | Aug 2016 to May 2017 | 1 | 278 | 95 m | $I_1$–$I_2$ |


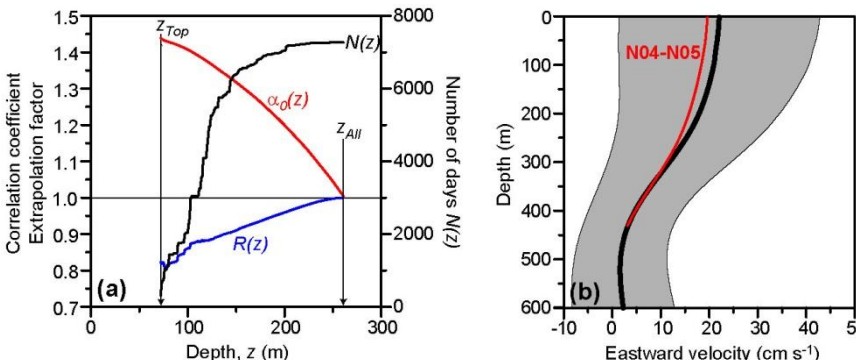

**Figure 3.** An illustration of the extrapolation of ADCP velocities using the eastward velocity at ADCP site NB as an example. **(a)** Depth variation of three parameters between the depth $z_{All}$, which is the shallowest depth with no error in daily averaged velocities and the depth $z_{Top}$, which is the shallowest depth with some good data: $N(z)$ is the number of days with good data at depth $z$. $\alpha_0(z)$ is the extrapolation factor at depth $z$, defined by Eq. (1). $R(z)$ is the correlation coefficient between $u(z,t)$ and $u(z_{All},t)$ where $u(z,t)$ is the eastward velocity at NB for depth $z$ and time $t$. **(b)** Vertical variation of the eastward velocity at ADCP site NB. The thick black curve shows the average extrapolated ADCP velocity profile with the grey area showing average ± one standard deviation. The red curve shows the average baroclinic velocity profile for standard CTD station interval N04 – N05, which includes ADCP site NB, adjusted so that it matches the average ADCP velocity at its deepest level.





This extrapolation method can only be used up to the Top depth. For the four ADCP sites on the IFR, we do not have the necessary additional information to extend the extrapolation. For these sites, the velocity for each day is extended unchanged up to the surface. For the ADCP sites on the N-section, in contrast, there are many near-synoptic CTD profiles on the standard stations during cruises along the section (Table 1). For each of these cruises, the geostrophic method may be

used to calculate the vertical variation of the eastward velocity in each interval between neighbouring standard stations. For most of the intervals, the eastward velocity typically has only small changes in the uppermost 100 m and the extrapolation factor may be modified to account for these (Hansen et al., 2019). With these modifications, all the daily ADCP profiles from the N-section have been extrapolated to the surface (Fig. 3b).

**2.3 Satellite-tracked drifter data**

Quality controlled data (1990–2018), interpolated to 6-hour intervals, from satellite-tracked drifters from the Global Drifter Program in the area (0°–30° W, 50°–65° N) are available from NOAAs Atlantic Oceanographic and Meteorological Laboratory (AOML) (http://www.aoml.noaa.gov/envids/gld/dirkrig/parttrk_spatial_temporal.php). The drifters have drogues at 15 m depth and only data with the drogue attached are used here**.**

**2.4 Sea level height from satellite altimetry**

Both the old and the new versions of the altimetry data were selected from the global gridded (0.25°x0.25°) sea level anomaly (SLA) field available from Copernicus Marine Environment Monitoring Service (CMEMS) (http://marine.copernicus.eu):

- **Old altimetry data set:** SEALEVEL_GLO_PHY_L4_REP_OBSERVATIONS_008_047
- **New altimetry data set:** SEALEVEL_GLO_PHY_L4_MY_008_047


From both of these data sets, daily SLA time series were selected for 8 grid points parallel to the N-section, here labelled $A_1$ to $A_8$, along 6.125° W from 62.125° N to 63.875° N (Fig. 2). We also use SLA data from the new data set for 10 grid points $I_1$ to $I_{10}$ along a line following the crest of the IFR (Fig. 2a) and we use gridded values for the Mean Dynamic Topography (MDT) associated with both data sets (Mulet et al., 2021).

**2.5 Statistical methods**

Correlations between two data sets are estimated by the Pearson correlation coefficient. To account for serial correlation in the data, the statistical significance of correlation coefficients is corrected by the modified Chelton method recommended by Pyper and Peterman (1998). Significance is indicated by asterisks: * means $p < 0.05$. ** means $p < 0.01$. *** means $p < 0.001$. No asterisk means $p > 0.05$.



For averages, the 95 % confidence limits are estimated as the standard errors multiplied by 1.96, corrected for serial correlation by replacing the sample size by the "equivalent sample size" (von Storch, 1999) calculated from the auto-correlation of the time series. Confidence limits for coefficients determined by linear regression are corrected similarly.

## 3 The accuracy of surface velocity anomalies derived from gridded altimetry

With the large data sets of ADCP and other in situ measurements along the N-section during the altimetry period, we have
the possibility to check how accurately surface velocity may be derived from altimetry data. This will be done using both the old and the new altimetry versions. For that purpose, we use altimetry data from grid points $A_1$ to $A_8$, which are along a line close to and parallel to the N-section (Fig. 2).

    The basic assumption is that, on sufficiently long time scales, geostrophic balance implies proportionality between surface velocity in a given direction and the slope of the sea surface perpendicular to that direction. For any given $k$ (=1,...,7),
the eastward surface ($z = 0$) velocity at time $t$, $U_k(0,t)$, horizontally averaged between altimetry points $A_k$ and $A_{k+1}$, should be proportional to the difference in absolute sea level height (SLH) between $A_k$ and $A_{k+1}$. The SLA value, $H_k(t)$, at grid point $A_k$ does not represent absolute SLH (above the geoid), but rather the anomaly from the MDT. The surface velocities, derived directly from SLA-differences between two grid points, are therefore also anomalies, but they may be converted to absolute velocities by adding a constant, which we will refer to as the ***"Altimetric offset"***, $U_k^0$, for each interval:


$$U_k(0,t) = \frac{g}{f \cdot L} \cdot [H_k(t) - H_{k+1}(t)] + U_k^0 \equiv \alpha_{Th} \cdot \Delta H_k(t) + U_k^0 \tag{2}$$

where $g$ and $f$ are gravity and Coriolis parameter, respectively, $L$ is the distance between the altimetry grid points and we have defined: $\Delta H_k(t) \equiv [H_k(t) - H_{k+1}(t)]$ as well as the coefficient $\alpha_{Th} \equiv g/(f \cdot L)$ according to geostrophic theory.
In order to check Eq. (2) by using ADCP data, we may replace $U_k(0,t)$ in the equation by the extrapolated surface velocity from an ADCP, $u(0,t)$, where we use lower case $u$ to emphasize that it is not horizontally averaged. This is compared with the SLA-difference, $\Delta H(t)$, for an altimetry interval that straddles the ADCP location. If Eq. (2) is to be a good approximation, there has to be a linear relationship between $u(0,t)$ and $\Delta H(t)$, which may be checked by calculating the correlation coefficient. Also, the coefficient, $\alpha_{Reg}$, determined by a regression analysis of Eq. (3), should have the theoretical
value: $\alpha_{Reg} = \alpha_{Th}$. Table 3 presents a first test of Eq. (2) by comparing 28-day averaged surface velocities from individual ADCP sites with SLA-differences on monthly time scales using both the old and the new altimetry data sets.

$$u(0,t) \cong \alpha_{Reg} \cdot \Delta H(t) + b \tag{3}$$




**Table 3.** Comparison between correlation and regression coefficients based on old and new altimetry data. "$R$" is the correlation coefficient with statistical significance (Sect. 2.5) between 28-day averaged values for eastward surface velocities from individual ADCP sites, $u(0,t)$, and differences in SLA values between the two neighbouring altimetry points that straddle the ADCP location, $\Delta H(t)$. "$\alpha_{Reg}$" and "$b$" are the regression coefficients in Eq. (3) with 95 % confidence limits. The theoretical value of $\alpha_{Th}$ is based on Eq. (2). "$N$" is the number of contiguous 28-day averaged values in each analysis.

| Site | $N$ | $R$ | | $\alpha_{Reg}$ (s$^{-1}$) | | $\alpha_{Th}$ (s$^{-1}$) | $b$ (cm s$^{-1}$) | |
| --- | --- | --- | --- | --- | --- | --- | --- | --- |
| | | Old | New | Old | New | | Old | New |
| NI | 12 | 0.42 | 0.67 | 2.4±3.8 | 2.8±2.7 | 2.72 | 12.4±3.0 | 12.7±3.0 |
| NA | 231 | 0.28*** | 0.39*** | 1.2±0.6 | 1.3±0.4 | 2.72 | 18.2±1.0 | 18.2±1.0 |
| NE | 95 | 0.78*** | 0.84*** | 5.0±0.8 | 4.1±0.6 | 2.72 | 24.3±1.4 | 24.2±1.2 |
| NB | 253 | 0.73*** | 0.77*** | 4.3±0.5 | 3.4±0.4 | 2.71 | 22.7±1.1 | 22.6±1.0 |
| NG | 167 | 0.61*** | 0.62*** | 3.1±0.6 | 2.3±0.5 | 2.71 | 12.6±1.3 | 12.5±1.3 |
| NC | 53 | 0.39** | 0.42** | 2.1±1.4 | 1.7±1.0 | 2.71 | 8.6±1.9 | 8.4±1.9 |
| NH | 12 | 0.65* | 0.85*** | 5.7±4.7 | 4.9±2.2 | 2.70 | 10.1±5.0 | 10.1±3.4 |

All of the correlation coefficients in Table 3 are higher when using the new rather than the old altimetry data. Even with the new data, most of the correlations are low, however, and the regression coefficients, $\alpha_{Reg}$, are in most cases different from the theoretical values, $\alpha_{Th}$. Here it must be taken into account that the velocity, $U_k(0,t)$, in Eq. (2) should be the horizontally averaged velocity for the whole interval between the two altimetry grid points, whereas the ADCP velocities are for the specific location of the ADCP.

For a more appropriate test of Eq. (2), we note that the interval $A_4$–$A_5$ includes two ADCP sites, NB and NG. We may therefore approximate the horizontally averaged eastward surface velocity, $U_4(0,t)$, in this interval by combining the ADCP velocities from the two sites. The simplest attempt would be a linear combination of the surface velocities from ADCP sites NB and NG:

$$U_4(0,t) = \beta_{NB} \cdot u_{NB}(0,t) + \beta_{NG} \cdot u_{NG}(0,t) \tag{4}$$

where we require the weighting factors to add up to one ($\beta_{NB} + \beta_{NG} = 1$) to indicate that each of the two ADCP sites represents a fraction of the altimetry interval. To determine the optimal combination of coefficients, we use a least-squares approach, varying $\beta_{NB}$ and $\beta_{NG}$ between 0 and 1 under the constraint above and minimizing the standard deviation of the residual:

$$Res(t) = \beta_{NB} \cdot u_{NB}(0,t) + \beta_{NG} \cdot u_{NG}(0,t) - \frac{g}{f \cdot L} \cdot \Delta H_4(t) \tag{5}$$

Once the weighting factors have been determined, the resulting time series, $U_4(0,t)$, can be correlated with $\Delta H_4(t)$ to check whether this improves the correspondence between ADCP-derived and altimetry-derived surface velocity. This was done using both the old and the new datasets (Table 4) and the correlation coefficients are now much higher than in Table 3, especially with the new altimetry data. A similar procedure may be carried out for the altimetry interval $A_3$–$A_4$, where there



are two ADCP sites, NA and NE. From Fig. 4, the surface velocity typically has a maximum between NA and NB, and NB is
quite close to the interval (Fig. 2b). We therefore approximate the horizontally averaged surface velocity in this interval,
$U_3(0,t)$, as a linear combination of surface velocities from these three ADCPs:

$$U_3(0,t) \cong \gamma_{NA} \cdot u_{NA}(0,t) + \gamma_{NE} \cdot u_{NE}(0,t) + \gamma_{NB} \cdot u_{NB}(0,t) \qquad (6)$$


where we again require that $\gamma_{NA} + \gamma_{NE} + \gamma_{NB} = 1$ and do a least squares analysis to determine the weighting factors. Also, for
this case the correlation coefficients are higher when using the new rather than the old altimetry data (Table 4).

**Table 4.** Weighting factors in Eq. (4) and Eq. (6) as well as correlation ($R$) and regression coefficients (with 95 % confidence limits)
between 28-day averaged values for eastward surface velocities generated from ADCP data and SLA-differences for the A3–A4 and A4–A5
intervals, respectively. $\alpha_{\mathrm{Reg}}$ is the coefficient in the regression equation $U(0,t) = \alpha_{\mathrm{Reg}} \cdot \Delta H(t) + b$. $a_{\mathrm{Th}}$ is the theoretical coefficient. "$N$" is the
number of contiguous 28-day periods for each analysis.

| | $U_3$ versus $\Delta H_3$ ($N$=94) | | | | | | $U_4$ versus $\Delta H_4$ ($N$=166) | | | | |
|---|---|---|---|---|---|---|---|---|---|---|---|
| | $\gamma_{NA}$ | $\gamma_{NE}$ | $\gamma_{NB}$ | $R$ | $\alpha_{\mathrm{Reg}}$ (s$^{-1}$) | $\alpha_{\mathrm{Th}}$ (s$^{-1}$) | $\beta_{NB}$ | $\beta_{NG}$ | $R$ | $\alpha_{\mathrm{Reg}}$ (s$^{-1}$) | $\alpha_{\mathrm{Th}}$ (s$^{-1}$) |
| Old altim.: | 0.31 | 0.32 | 0.37 | 0.82*** | 3.69±0.53 | 2.72 | 0.51 | 0.49 | 0.89*** | 3.69±0.30 | 2.71 |
| New altim.: | 0.33 | 0.34 | 0.33 | 0.86*** | 2.87±0.36 | 2.72 | 0.53 | 0.47 | 0.92*** | 2.89±0.20 | 2.71 |

Another important result in Table 4 is that the regression coefficients between horizontally averaged ADCP-velocity
and SLA-difference agree with the theoretical values within confidence limits for the new altimetry data, but not for the old
data.

## 4 The surface circulation between Iceland and the Faroes

In geostrophic balance, the average surface velocity is parallel to the MDT-isolines, and the speed of the flow is higher the
closer the isolines are. A map of the MDT (Fig. 4) should therefore give a picture of the surface circulation and this picture
indicates two inflow branches: an "***Icelandic branch***" over the northern end of the IFR and a "***Faroese branch***" over the
southern end, but no consistent inflow across the middle of the ridge.

This picture is consistent with the extrapolated surface velocities at the eleven ADCP sites (Fig. 4). The ADCP at site
IW was located on the Icelandic flank of the "Western Valley" (Fig. 2a), and it shows a strong inflow with average surface
velocity exceeding 20 cm s$^{-1}$ in magnitude. According to the MDT, a part of this inflow continues directly towards the N-
section, but another part circles back onto the ridge before returning eastwards in a form of "***retroflection***". As part of this
process, the MDT indicates a "***recirculation***" over the northernmost bank on the IFR, indicated by "RB" in Fig. 4. Over the
south-eastern half of the IFR, both the MDT and the ADCPs indicate average inflow in the surface layer. The water that has
crossed the IFR in the surface layer continues towards the N-section, where it is focused into a narrow current, the Faroe
Current, with a high-velocity core located close to ADCP site NE on average (Fig. 4).




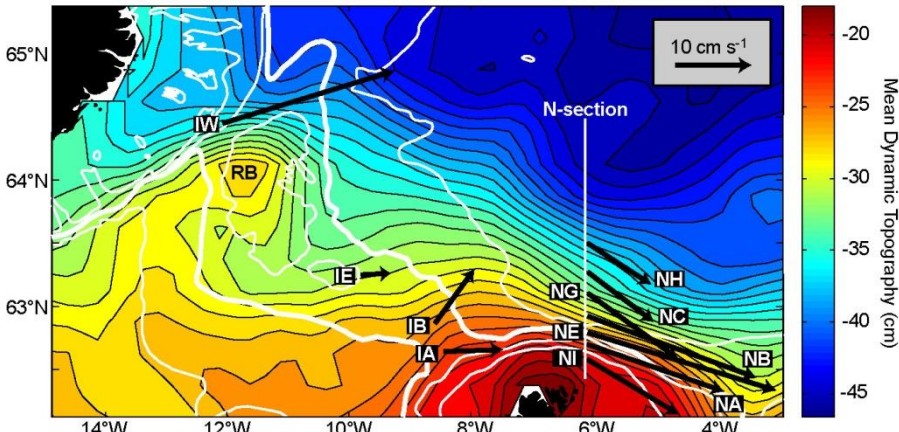

**Figure 4.** The surface circulation between Iceland and the Faroes. The background colours show the MDT (new altimetry data set, Mulet et al., 2021). Average surface velocities at the eleven ADCP sites are shown with arrows that start at the site and have lengths according to the scale in the right top corner (extrapolated data, see Sect. 3). The "RB" indicates the Rosengarten Bank, over which there is a recirculation region according to the MDT. White lines show isobaths for 200 m, 400 m, 500 m, and 1000 m, with the 500 m isobath thicker than the others.

The circulation map based on the MDT and ADCP data (Fig. 4) is largely consistent with the tracks shown by the satellite-tracked drifters (Fig. 5a). Drifters have passed over almost every part of the ridge with no apparent structure in the pathways (little topographic steering). This is somewhat misleading, however, as indicated in Fig. 5b. This figure focuses on drifters passing through the Western Valley and they tend to follow a narrow path over the Icelandic slope. A total of twelve drifters passed through the Western Valley southeast of the 200 m isobath. Eleven of them kept within a corridor around 10 km wide located above ADCP site IW. This is where Fig. 4 shows a strong inflow velocity in the surface, and it indicates that there may be a fairly narrow high-speed jet over the Icelandic slope.

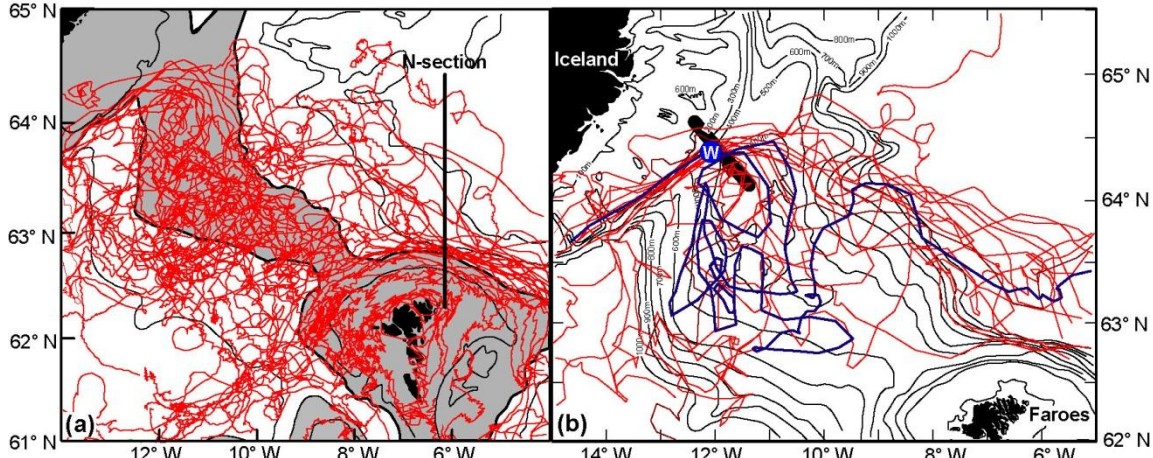

**Figure 5. (a)** Daily averaged tracks of drifters that crossed the IFR from the Iceland Basin to the Norwegian Basin in the 1991–2018 period (red traces). The shaded area is shallower than 500 m. **(b)** Tracks of 16 drifters that crossed the Altimetry line over the IFR (Fig. 2a) the first time northwest of altimetry point $I_3$ (over the thick black line). The thick blue trace shows one specific drifter track that has been enhanced to illustrate retroflection and recirculation. The blue circle, labelled W indicates ADCP site IW.





Some of the drifters in Fig. 5b are seen to originate from southerly parts of the eastern Iceland Basin. Hence, the jet is
not solely fed from water over the south Icelandic slope. East of site IW, the jet seems to lose the topographical steering,
turning towards southeast. Many of these drifters are seen to return back onto the IFR as more clearly illustrated by the blue
trace in Fig. 5b. This verifies that both retroflection and recirculation do indeed occur over the IFR.

## 5 The inflow across the IFR

From Fig. 4 and Fig. 5, it is clear that the inflow across the IFR is not as simple as some of the early maps (e.g., Meincke,
1983; Hansen and Østerhus, 2000) indicated. By combining the ADCP observations with satellite altimetry, the details may
be clarified.

### 5.1 Depth-variation of the inflow across the IFR

The data acquired at each of the ADCP sites on the IFR (Table 2) may be used to estimate average velocities and their
variations at various depths at these four sites. We define the "***cross-ridge velocity***" as the velocity component perpendicular
to the altimetry line following the ridge crest (red line on Fig. 2a), and directed towards the Norwegian Basin. Periods with
positive cross-ridge velocity are termed "inflow", whereas negative velocity is termed "outflow" (even though this water
may later turn back towards the Norwegian Basin).

According to Fig. 4, all the ADCP sites on the IFR had positive cross-ridge velocities in the surface, on average. Going
from the surface towards the bottom, the average (over the deployment period) cross-ridge velocity decreased at all the
ADCP sites on the IFR, as illustrated in Fig. 6. In this figure, each of the profiles has a layer just beneath the surface where
the velocity appears not to change with depth. This is due to the method used over the IFR for extrapolating ADCP-
velocities towards the surface (Sect. 2.2). Its effect is most notable for site IW, where the average surface velocity is likely
underestimated by the extrapolation method.

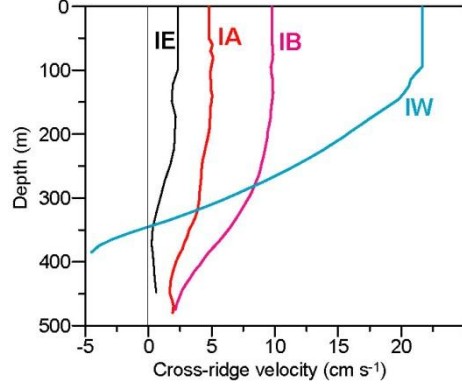

**Figure 6.** Average profiles of cross-ridge velocity for each of the four ADCP deployments on the IFR.



Except for site IW, the average cross-ridge velocity varies little with depth in the uppermost 200–300 m, below which it weakens. Close to the bottom, the cross-ridge velocity is positive at all ADCP sites except for IW where it is negative, indicating overflow (Hansen et al., 2018).

## 5.2 Temporal variations of the inflow across the IFR

On weekly time scales, periods with outflow occurred at all the ADCP sites (Fig. 7). They document that the Atlantic water flow across any one location on the ridge is not a continuous process, but involves considerable motion back and forth, as also indicated by the drifters (Fig. 5). Since the ADCP sites are all close to the altimetry line following the crest ($I_1$ to $I_{10}$), we may correlate the cross-ridge surface velocity with SLA-differences across intervals between neighbouring altimetry grid points (Table 5) where we will only use the new altimetry data. ADCP sites IB and IW were located close to the centre of one of the intervals and they exhibit highly significant positive correlations. Sites IA and IE, in contrast, were located away from the centres and their correlations are lower. Table 5 also includes some negative correlations when ADCP velocities are correlated with SLA-differences across intervals that do not contain the ADCP site. This will be addressed in Sect. 5.5 and Sect. 7.2.

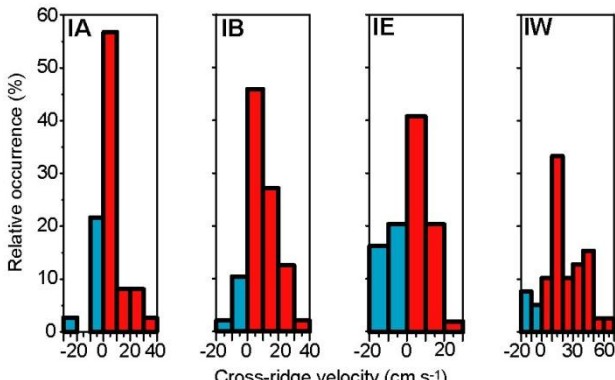

**Figure 7.** Histograms of 7-day averaged cross-ridge surface velocity for each of the ADCP sites on the IFR. Cyan bars show negative and red bars positive cross-ridge surface velocities.

**Table 5.** Correlation coefficients between 7-day averaged cross-ridge surface velocity at each of the ADCP sites on the IFR (Fig. 2a) and the SLA-difference between neighbouring altimetry grid points (second point minus first point) on the altimetry line following the crest of the ridge (Fig. 2a). Bold values indicate that the ADCP was located within the interval or in a neighbouring interval, but close to the separating grid point.

| Interval: | $I_1$-$I_2$ | $I_2$-$I_3$ | $I_3$-$I_4$ | $I_4$-$I_5$ | $I_5$-$I_6$ | $I_6$-$I_7$ | $I_7$-$I_8$ | $I_8$-$I_9$ | $I_9$-$I_{10}$ |
|---|---|---|---|---|---|---|---|---|---|
| Site IA: | -0.08 | -0.02 | 0.17 | 0.01 | -0.24 | 0.06 | -0.49** | **0.25** | **0.45*** |
| Site IB: | 0.23 | 0.05 | -0.36* | -0.14 | 0.26 | -0.12 | -0.29 | **0.63*** | -0.43** |
| Site IE: | 0.23 | 0.16 | -0.17 | -0.10 | -0.19 | **0.37** | **0.09** | -0.36* | -0.10 |
| Site IW: | **0.88*** | -0.11 | -0.62*** | -0.35* | 0.18 | -0.24 | 0.02 | 0.12 | -0.06 |



## 5.3 The "equivalent width" of a surface current

Most of the correlation coefficients in Table 5 are quite low, although some of them are highly significant. For most of the ADCP sites, the correlation coefficient increases substantially when averaging over 28 rather than 7 days and the correlation may also increase by choosing a wider altimetry interval (Table 6).

**Table 6.** Correlation coefficients, $R$, and regression coefficients, $\alpha_{Reg}$, with 95 % confidence limits for Eq. (3) where data have been averaged over 7 and 28 days, respectively, before analysis. The last three columns list the theoretical regression coefficient, $\alpha_{Th}$, the width of the interval, $L$, and the Equivalent width, $L_{Eq}$, as defined in the text.

| ADCP Site | Altim. interv. | $R$ 7-day | $R$ 28-day | $\alpha_{Reg}(s^{-1})$ 7-day | $\alpha_{Reg}(s^{-1})$ 28-day | $\alpha_{Th}(s^{-1})$ | Width (km) $L$ | Width (km) $L_{Eq}$ |
|---|---|---|---|---|---|---|---|---|
| IA | $I_9$-$I_{10}$ | 0.45* | 0.42 | 1.6±1.1 | 1.2±2.5 | 1.60 | 48 | 48 |
| IB | $I_8$-$I_9$ | 0.63*** | 0.82** | 1.6±0.6 | 2.0±1.0 | 1.60 | 47 | 47 |
| IE | $I_6$-$I_8$ | 0.38* | 0.89*** | 0.8±0.6 | 1.3±0.5 | 0.80 | 94 | 94 |
| IW | $I_1$-$I_2$ | 0.88*** | 0.97*** | 6.1±1.2 | 6.4±1.8 | 2.03 | 37 | 12 |
| IW | $I_1$-$I_3$ | 0.80*** | 0.88** | 5.5±1.4 | 5.7±2.9 | 1.01 | 74 | 14 |

Except for site IA, Table 6 indicates that on monthly time scales, the ADCP-derived surface velocity does represent the horizontally averaged velocity well. From the results in Sect. 3, we would then expect the regression coefficient $\alpha_{Reg}$ to equal the theoretical value, $\alpha_{Th} = g/(f \cdot L)$, in Eq. (3) when using the new SLA data. For most of the ADCP sites in Table 6, $\alpha_{Reg}$ is equal to $\alpha_{Th}$ within the (wide) confidence limits, but not for site IW.

The ADCP at site IW was located close to the middle of interval $I_1$–$I_2$ (Fig. 8) and for 28-day averaged data, the correlation coefficient is very close to 1 (Table 6). Nevertheless, the regression coefficient, $\alpha_{Reg}$, is three times the theoretical value, $\alpha_{Th}$. The validity of the regression coefficient for site IW depends on the extrapolation of ADCP velocity to the surface, but Fig. 6 indicates that the extrapolation method has underestimated the real surface velocity at ADCP site IW, rather than the opposite. Thus, errors in the extrapolation to surface values cannot explain the large difference between $\alpha_{Reg}$ and $\alpha_{Th}$ for this site.

This apparent discrepancy may be explained by the narrow high-speed surface jet that was indicated by the drifter data (Fig. 5b). If this jet contains more or less all the surface flow between $I_1$ and $I_2$, while being topographically locked to a fixed location above IW, and having a fixed width (Fig. 8), then the SLH change will be proportional to the surface velocity measured by the ADCP. This would explain the high correlation and the regression coefficient will be larger than the theoretical value as long as the jet is narrower than the width of the altimetry interval.

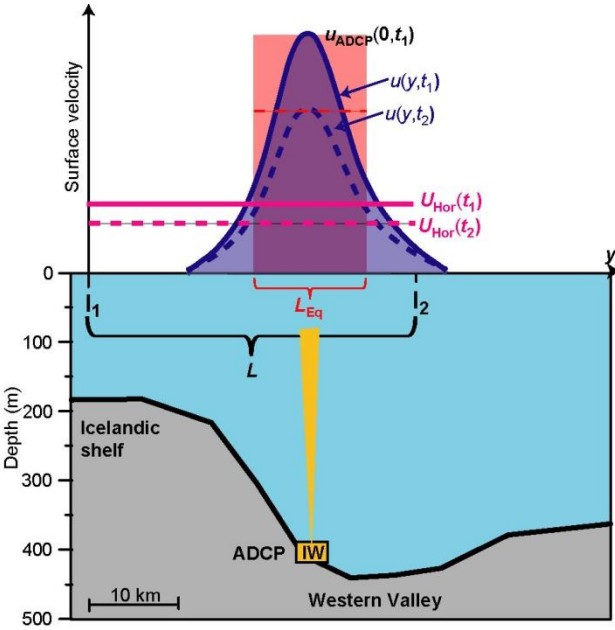


**Figure 8.** Interpretation of the ADCP observations at site IW and their relationship with altimetry data. The bottom part of the figure shows a section going through altimetry points $I_1$ and $I_2$, on which the ADCP was deployed. The thick blue curves in the top part of the figure show an hypothetical horizontal variation of the cross-ridge surface velocity, $u(y,t)$, at two times: $t_1$ (continuous) and $t_2$ (dashed). The two horizontal magenta lines show the horizontally averaged velocity, $U_{Hor}(t)$, for the same times. The equivalent width of the jet, $L_{Eq}$,

is defined such that the product $L_{Eq} \cdot u_{ADCP}(0,t)$ (the red area) is equal to the horizontal integral of $u(y,t)$ (the blue area).

As illustrated in Fig. 8, we can define a parameter, the "***equivalent width***", $L_{Eq}$, which should be a good estimate of the width of the jet and may be derived from the width of the altimetry interval, $L$: $L_{Eq} \approx (\alpha_{Th}/\alpha_{Reg}) \cdot L$. For ADCP IW together with interval $I_1$–$I_2$, we find $L_{Eq} \approx$ 12 km (Table 6). This is of a similar magnitude as the width estimated from the drifters

(Fig. 5b) and also similar to the baroclinic Rossby radius in the region, in support of this interpretation. Although the width of this jet is only one third of the width of $I_1$–$I_2$, its surface velocity is apparently sufficient to dominate the sea level tilt across the interval. And it appears to dominate the tilt across the wider interval $I_1$–$I_3$ as well, as indicated by the high correlation in the bottom row of Table 6.

Similar arguments may be used for the other sites as long as the correlations in Table 6 remain high. This is the case to

some extent for 28-day averaged data, especially for IB and IE. In contrast to IW, the other three sites do not show disagreement between $\alpha_{Reg}$ and $\alpha_{Th}$ within the (wide) confidence limits (Table 6). For these sites, $L_{Eq}$ has therefore been set equal to $L$.

### 5.4 Volume transport of inflow across the IFR

The main reason for introducing the equivalent width is that this parameter can help us to make some rough estimates of

volume transport across the different parts of the IFR. To do this, the ridge is split into five intervals, $k = 1,...,5$, delimited by



altimetry grid points as listed in Table 7. Each of the intervals is represented by one of the ADCPs, except for the second interval, $I_3$–$I_6$, which is included for completeness. For the other four intervals, the average volume transport through the interval may be estimated as the equivalent width times the vertical integral of the average cross-ridge velocity measured by the ADCP in the interval, $\langle u_k(z) \rangle$. Since we are only interested in the inflow, we only integrate down to the depth, $z = z_0$,

where the average cross-ridge velocity becomes zero:

$$\langle Q_k \rangle \cong L_{Eq,k} \cdot \int_{z=0}^{z_0} \langle u_k(z) \rangle \, dz \equiv L_{Eq,k} \cdot D_{Eq,k} \cdot \langle u_k(0) \rangle \tag{7}$$

where the last expression may be seen as a definition of the "***equivalent depth***", $D_{Eq,k}$, for interval $k$. If the flow were fully

barotropic, this parameter would be the depth needed to give the same volume transport as the real flow according to the average ADCP velocity profile. Consistent with Fig. 6, all the ADCP sites in Table 7 have equivalent depths that are smaller than the bottom depth at the site. The average transport estimate for each interval is listed in the bottom row of Table 7. They add up to 4.0 Sv, but the transport between $I_3$ and $I_6$ is probably negative (Fig. 4), which would make the total sum somewhat smaller.


**Table 7.** Average volume transport of inflow across the IFR split into five intervals.

| Interval $k$: | 1 | 2 | 3 | 4 | 5 |
|---|---|---|---|---|---|
| Altim. int.: | $I_1$-$I_3$ | $I_3$-$I_6$ | $I_6$-$I_8$ | $I_8$-$I_9$ | $I_9$-$I_{10}$ |
| ADCP: | IW | - | IE | IB | IA |
| $\langle u_k(0) \rangle$ (cm s$^{-1}$): | 21.6 | - | 2.4 | 9.8 | 4.8 |
| $L_{Eq,k}$ (km): | 14 | - | 94 | 47 | 48 |
| $D_{Eq,k}$ (m): | 247 | - | 290 | 379 | 397 |
| $\langle Q_k \rangle$ (Sv): | 0.7 | - | 0.7 | 1.7 | 0.9 |

Attempts to construct time series of volume transport through the various intervals in Table 7 were found to be too sensitive to the required approximations for most of the intervals. For the Icelandic branch, however, the correlations in

Table 6 are so high that it seems reasonable to look for temporal variations in the form:

$$Q_I(t) = \langle Q_1 \rangle + \frac{g \cdot D_{Eq,1}}{f} \cdot \Delta H_{IB}(t) \tag{8}$$

where $\Delta H_{IB}(t)$ is the SLA-difference across the Icelandic branch, i.e., the difference between $I_1$ and $I_3$, and $D_{Eq,1}$ is the

equivalent depth for ADCP site IW (247 m). Monthly averaged values for $Q_I(t)$ (Fig. 9a) vary considerably with a few months even showing negative transport. This is consistent with the extrapolated surface velocities at ADCP site IW (Fig. 7). Over the altimetry period, the volume transport of the Icelandic branch had a consistent seasonal variation with the transport in June being only half of that in February–March, on average. The seasonal variation is also seen in the cross-ridge surface velocity through interval $I_1$–$I_3$, as shown by the cyan curve in Fig. 9b. This figure also illustrates the seasonal velocity





variation through the "recirculation region (I₃–I₆)" and the Faroese branch (I₆–I₁₀) as well as the whole width of the IFR (I₁–I₁₀).

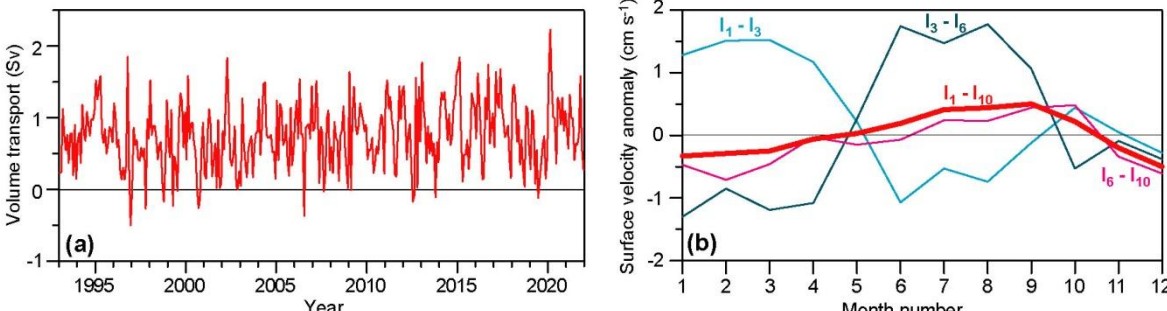

**Figure 9. (a)** Monthly average volume transport of the Icelandic branch as estimated by Eq. (8). **(b)** Seasonal variation of the cross-ridge
surface velocity anomaly, horizontally averaged across the whole ridge (I₁–I₁₀, thick red curve) and across three altimetry intervals (thin curves). The velocity anomaly for each interval is directed perpendicular to the line connecting the interval end points and directed towards the Norwegian Basin.

## 5.5 Retroflection and recirculation of the Icelandic branch

In addition to the positive correlation coefficient between surface velocity at site IW and SLA-difference across I₁–I₂, Table
5 also shows a highly significant negative correlation (-0.62***) between this velocity and SLA-difference across I₃–I₄. This indicates that the recirculation around the northernmost bank on the IFR varies with the strength of the Icelandic branch. This link is further explored in Fig. 10, which shows the anomalous slope of the sea surface and surface flow anomaly associated with strong flow in the Icelandic branch.

More precisely, Fig. 10 shows the coefficient $a_{i,j}$ in Eq. (9). Here, $H_{i,j}(t)$ is the SLA value at point (i,j) (i=1,...,N,
j=1,...,M) in a subset of the altimetry grid that covers the area in Fig. 10. The two points I₁ and I₂ are located at (i1,j1) and (i2,j2), respectively, in this grid. The bracket on the right-hand side of Eq. (9) is therefore proportional to the strength (surface velocity) of the Icelandic branch at time $t$. The bracket on the left-hand side of the equation is the SLA value in each grid point at time $t$ minus the spatially averaged SLA for the whole region at this time. The reason for subtracting this average is to reduce the variability induced by long-term and seasonal sea level variations so that the figure more directly
represents the anomalous slope of the sea surface, which is related to velocity through geostrophy. By this choice, the figure also becomes independent of an accurate MDT.

$$\left[ H_{i,j}(t) - \frac{1}{N \cdot M} \cdot \sum_{n,m} H_{n,m}(t) \right] \cong a_{i,j} \cdot \left[ H_{i2,j2}(t) - H_{i1,j1}(t) \right] + b_{i,j} \tag{9}$$



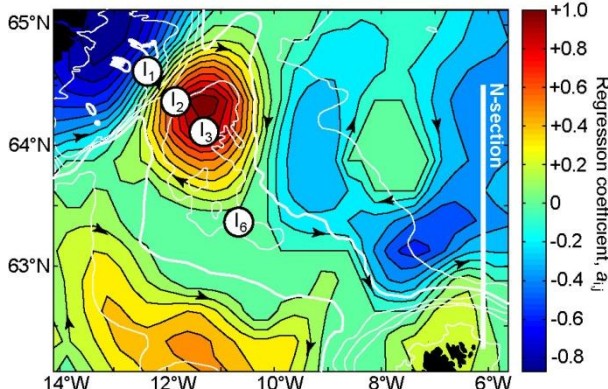

**Figure 10.** The surface flow anomaly associated with a strong inflow through the Western Valley. The colours show the regression coefficient, $a_{i,j}$, in Eq. (9) with 7-day averaged data throughout the altimetry period. Arrowheads indicate the anomalous flow direction. Grid points $I_1$ and $I_2$, with coordinates (i1,j1) and (i2,j2), respectively, are indicated by circles, as are grid points $I_3$ and $I_6$. If the correlation coefficient was not significantly different from zero at the 0.001 level (p>0.001), the regression coefficient was set to zero.

# 6 Monitoring volume transport of Atlantic water through the N-section

The monitoring system described by Hansen et al. (2015) was designed to generate monthly averaged transport values in several steps. In the following, these are discussed as well as the dependence of volume transport estimate on in situ observations.

## 6.1 Determining absolute surface velocities on the N-section

As discussed in Sect. 3, surface velocity anomalies may be accurately derived from SLA data on monthly time scales. To generate ***absolute*** eastward surface velocities, we need in addition to determine the altimetric offsets, $U_k^0$, defined in Eq. (2), for $k = 2$ to 7. Around 70 % of the Atlantic water transport passes between $A_3$ and $A_5$, on average. The values for $U_3^0$ and $U_4^0$ are therefore especially important. From the analysis in Sect. 3, they may be determined with uncertainties of 1.0 cm s$^{-1}$ and 0.7 cm s$^{-1}$, respectively, i.e., less than 5 %.

For the other intervals, the values for "$b$" in Table 3 may be used, but they have high uncertainties as illustrated by the error bars in Fig. 11 and they are not based on horizontal averages. Over the northern part of the section, there are, however, many CTD profiles (Table 1), from which the average eastward velocity variation with depth can be determined for each interval between standard stations by using the classical geostrophic method. When this is combined with current meter measurements at depth, alternative estimates for $U_5^0$, $U_6^0$, and $U_7^0$ may be derived (blue lines in Fig. 11), as detailed in Hansen et al. (2019).





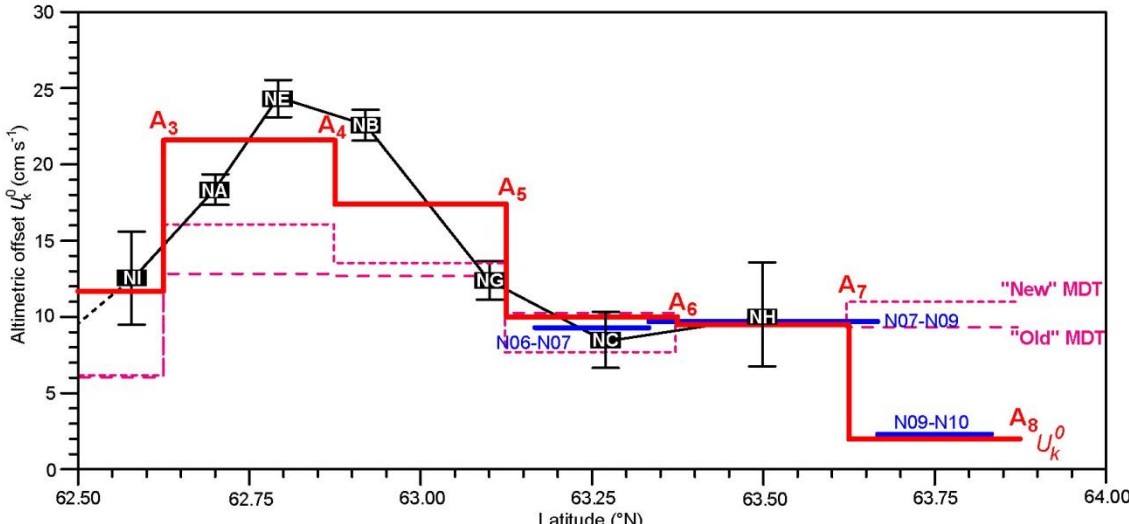

**Figure 11.** Optimized values for the Altimetric offset in each altimetry interval are shown by the thick continuous red line. Black rectangles with ADCP site names indicate $U_\mathrm{k}^0$ values with error bars indicating 95 % confidence limits for individual ADCP sites derived from the new altimetry data set (Table 3). Blue lines indicate $U_\mathrm{k}^0$ values derived from CTD data and measurements of deep currents (see Hansen et al., 2019). Dashed magenta lines show the values for $U_\mathrm{k}^0$ based on the MDT from the old and the new (Mulet et al., 2021) data sets.


As seen in Fig. 11, the ADCP-based and the CTD-based estimates agree well and may be combined to give optimized values for $U_\mathrm{k}^0$ as illustrated by the thick red line in Fig. 11. Except for the northernmost part of the section, with little

Atlantic water, alternative estimates of $U_\mathrm{k}^0$, based on the MDT (dashed lines in Fig. 11) would give too low surface velocities. Errors in the $U_\mathrm{k}^0$ values will introduce a bias to the transport time series and an error in the average transport. Combining the uncertainties in Fig. 11, this bias should not exceed 0.25 Sv, which is half the quoted uncertainty of the average volume transport (Hansen et al., 2015).

### 6.2 Vertical variation and integration of cross-sectional velocity on the N-section

Once the eastward surface velocity has been determined, Eq. (A2) allows calculation of eastward velocity at any given depth. This equation is based on the approximation that the eastward velocity at a given depth is proportional to the eastward surface velocity at the same location. The proportionality factor for each altimetry interval, month, and depth has been derived from the ADCPs within the interval (Hansen et al., 2019). This approximation must be expected to become less accurate with increasing depth, but the velocity also tends to decrease with increasing depth. Thus, the vertical sum of

velocities, needed for transport calculation, might not be very sensitive to the approximation. This can be checked by correlating eastward surface velocities for each of the ADCP sites, $u(0,t)$, with the vertically integrated eastward velocity ($S_\mathrm{ADCP}$) down to the average depth of Atlantic water (bottom or 4 °C isotherm), $D_\mathrm{A}$:



$$S_{\text{ADCP}}(t) = \sum_{z=1}^{D_A} u(z,t) \tag{10}$$


**Table 8.** Average depth of the Atlantic layer ($D_A$) at the ADCP sites, number of 28-day averaged values ($N$) at each site, and correlation coefficient ($R$) between eastward surface velocity and integrated eastward velocity ($S_{\text{ADCP}}$) down to depth $D_A$, Eq. (10).

| Site: | NI | NA | NE | NB | NG | NC | NH |
|---|---|---|---|---|---|---|---|
| $D_A$ (m): | 156 | 300 | 428 | 362 | 301 | 255 | 151 |
| $N$: | 12 | 231 | 95 | 253 | 167 | 53 | 12 |
| $R$: | 0.969* | 0.898*** | 0.973*** | 0.986*** | 0.988*** | 0.989*** | 0.998*** |

For most of the ADCP sites, the correlation coefficients in Table 8 are very high. The lowest value is for site NA, but
this low value may be misleading, because the calculations for Table 8 were made without distinguishing between months.
As discussed by Hansen et al. (2019), the velocity profile at NA has a strong seasonal variation. This has been taken into
account when generating the proportionality factors for each interval and month in Eq. (A2).

**6.3 Determination of Atlantic water extent on the N-section**

A number of different types of in situ instruments have provided time series with information on Atlantic water extent: CTD,
PIES, ADCP temperature sensors (Sect. 2). The CTD profiles are, however, snapshots and the other two types of instrument
have only been active at specific locations and during limited periods. The only observations that have continuous coverage
during the whole of the altimetry period are the altimetry data themselves. It is therefore essential to evaluate how accurately
temporal variations in the Atlantic water extent can be determined from altimetry.

For the transport calculations, the most sensitive extent-parameter is the deep boundary along the section, and one may
wonder why the variations of this boundary should be related to the altimetry data. The answer is that the hydrographic fields
are linked to the velocity field (Hátún et al., 2004) through a kind of geostrophic adjustment. Apparently, there is a rapid
adjustment between barotropic (sea level) and baroclinic (density field) variations. To demonstrate this, the pressure, $P(t)$, at
time $t$ at a given point in the ocean may be split into three contributions: a constant, $P_0$, a barotropic pressure anomaly, $P_T(t)$,
and a baroclinic pressure anomaly, $P_C(t)$:


$$P(t) \equiv P_0 + P_T(t) + P_C(t) \equiv g \cdot \rho_0 \cdot D + g \cdot \rho_0 \cdot h(t) + g \cdot \int_{z=0}^{D} [\rho(z,t) - \rho_0]\, dz \tag{11}$$

where $z$ is the vertical coordinate (positive downwards from a fixed level), $D$ the depth of the point below that level, $h(t)$ the
height of the sea surface directly above the point at time $t$, $\rho(z,t)$ the density, and $\rho_0 = 1027.3$ kg m$^{-3}$ is a typical surface
density on the N-section. To demonstrate the adjustment process, we have calculated $P_C(t)$ at 400 m depth for all CTD
profiles 1996–2019 from the deep standard stations on the N-section and correlated these values with $P_T(t)$ derived from
SLA-values for the same day with a lag varying between -30 and +30 days. As demonstrated in Table 9, there is a rapid
adjustment (by vertical displacement of isopycnals) with lag no more than a day. If sea level changes at a certain point on the



section, the density field apparently adjusts within a day, partially compensating for the barotropic anomaly change. From

the regression analysis, the compensation in terms of pressure is between 66 % and 75 % on stations N04 to N08, but

decreases to less than 40 % at N10.

**Table 9.** Correlation and regression coefficients between baroclinic and barotropic pressure anomaly, $P_C(t)=a_{Lag}\cdot P_T(t-Lag)+constant$, at 400 m depth on standard stations N04 to N10. "$N$" is number of CTD profiles, "$R_0$" is the correlation coefficient with $Lag = 0$ and "$a_0$" the

corresponding regression coefficient. "$Lag_m$" is the lag (in days) that gives maximum absolute correlation, which is "$R_m$", and "$a_m$" is the corresponding regression coefficient.

| Stat. | $N$ | $R_0$ | $a_0$ | $Lag_m$ | $R_m$ | $a_m$ |
|---|---|---|---|---|---|---|
| N04 | 102 | -0.81*** | -0.66 | -1 | -0.82 | -0.67 |
| N05 | 100 | -0.85*** | -0.75 | 0 | -0.85 | -0.75 |
| N06 | 91 | -0.82*** | -0.68 | -1 | -0.83 | -0.69 |
| N07 | 95 | -0.87*** | -0.69 | -1 | -0.88 | -0.69 |
| N08 | 92 | -0.88*** | -0.66 | 0 | -0.88 | -0.66 |
| N09 | 91 | -0.84*** | -0.51 | 1 | -0.84 | -0.51 |
| N10 | 90 | -0.77*** | -0.39 | 0 | -0.77 | -0.39 |

By definition, the Atlantic water extends to the bottom or the 4 °C isotherm, slightly modified by variations of Atlantic

water temperature (Appendix A) and the high correlations in Table 9 motivate why the depth of this isotherm may be related

to sea level height and hence altimetry. The algorithm for determining the isotherm depth, $D_j(t)$, at each standard station, $N_j$,

was determined from the CTD data for N04 to N10 by multiple regression analysis (Hansen et al., 2020). Table 10 verifies

that the algorithms explain a considerable fraction of the variance for most of the stations, especially when using the new

altimetry data.

**Table 10.** All but the last columns list the fraction ($R^2$) of the variance of the 4 °C isotherm depth, $D_j(t)$, as observed by CTD that is explained by Eq. (A3) using both the old and the new altimetry data. The last column lists explained variance of $PcS_1(t)$ by Eq. (A6).

| | Explained variance of the 4 °C isotherm depth, $D_j(t)$ | | | | | | | $PcS_1(t)$ |
|---|---|---|---|---|---|---|---|---|
| | N04 | N05 | N06 | N07 | N08 | N09 | N10 | |
| Old altimetry: | 0.31 | 0.62 | 0.58 | 0.66 | 0.63 | 0.56 | 0.54 | 0.58 |
| New altimetry: | 0.35 | 0.70 | 0.67 | 0.71 | 0.69 | 0.58 | 0.56 | 0.60 |

Since a CTD profile is a snapshot, the CTD-based isotherm depths include variations on time scales of days and even

shorter. These short-term variations may be smoothed by using the PIES observations. The two-way travel time measured by

a PIES depends on sound velocity, which again depends on temperature (and salinity and pressure) in a well known manner.

It is therefore conceivable, that the PIES can provide estimates of isotherm depth. This is verified in Fig. 12 where we have

calculated (two-way) travel time for each individual CTD profile at the two standard stations where the PIES were located

and compared it with the 4 °C isotherm depth determined from the same profile. The fits shown by the curve in each of the

panels allows the calculation of isotherm depth from travel time with a Root Mean Square error less than 30 m.




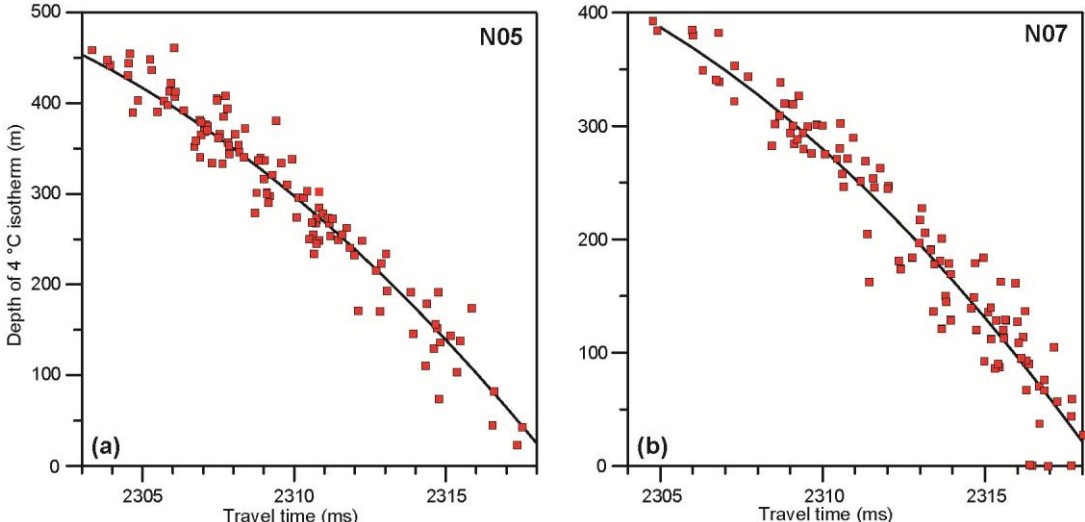

**Figure 12.** Depth of the 4 °C isotherm plotted against calculated travel time for sites N05 and N07 assuming a bottom depth of 1695 m. Each red square represents a CTD profile. Continuous lines indicate the fits.

Using these fits together with the travel time measurements of the two PIES, we can generate 28-day averaged values for the depth of the 4 °C isotherm and compare these values with the isotherm depths that are produced by Eq. (A3) with 28-day averaged altimetry (Table 11). Once again, the explained variances ($R^2$) are higher when using the new rather than the old altimetry data, but they are also considerably higher in Table 11 than in Table 10 and from the values for "Avg" in Table 11, there is no appreciable bias induced.


**Table 11.** The correspondence between 28-day averaged depths of the 4 °C isotherm for stations N05 and N07 as observed by the PIES (fits in Fig. 12) and as simulated by the expressions derived from the CTD data at the stations using both the old and the new altimetry data and coefficients. "$R^2$" is the variance explained by the fit. "Std" and "Avg" are the standard deviation and average of the difference (observed – simulated), respectively.

|                | N05    |       |      | N07    |       |      |
|----------------|--------|-------|------|--------|-------|------|
|                | $R^2$  | Std   | Avg  | $R^2$  | Std   | Avg  |
| Old altimetry: | 0.77   | 29 m  | 1 m  | 0.79   | 24 m  | -2 m |
| New altimetry: | 0.79   | 29 m  | -6 m | 0.84   | 21 m  | -1 m |


       From Table 10 it is clear that for standard station N04, the 4 °C isotherm depth at station N04 is not very accurately estimated by Eq. (A3). In periods when the bottom temperature at site NE has been measured, an improved estimate of this depth may be obtained by Eq. (A4). With the old altimetry data, the explained variance became $R^2 = 0.66$. With the new altimetry data, this again increases to $R^2 = 0.71$.

The final stage in determining the Atlantic water extent is to obtain an estimate of its northern boundary, which is based on salinity rather than temperature because of the seasonal warming of the surface layer (Fig. 2b). The last column in Table





10 shows that the explained variance of $PcS_1(t)$ increased from 0.58 to 0.60 when going from the old to the new altimetry data in Eq. (A6).

### 6.4 The dependence of transport accuracy on in situ observations

From the preceding results, the depth of the Atlantic water along the section may be estimated with fairly high accuracy even in periods without in situ observations of isotherm depth. This allows calculation of volume transport also in these periods, but presumably with less accuracy. To estimate the uncertainty induced by lack of in situ observations we have calculated time series of volume transport with and without these observations and compared them as indicated in Fig. 13a. The red squares in the figure are for 19 months, during which PIES were deployed at N05 and N07. From the PIES data, monthly
averaged isotherm depth can be generated for these two stations and for station N06 by interpolation (Hansen et al., 2020). The cyan squares, similarly, are for 115 months with bottom temperature measurements at site NE (Fig. 2b), which allow monthly averaged isotherm depth to be calculated at station N04 with higher accuracy (Eq. (A4) and Sect. 6.3).

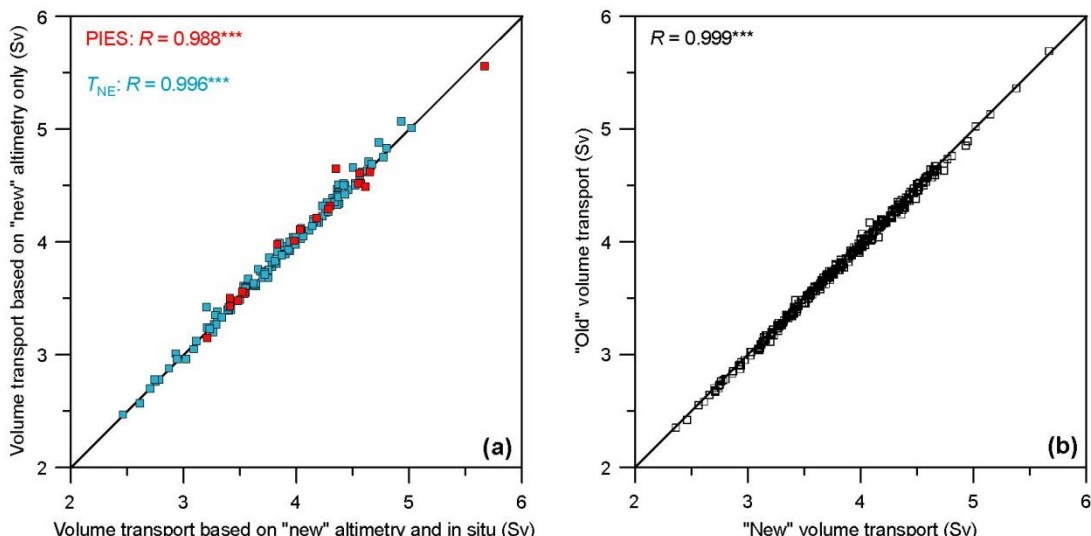

**Figure 13. (a)** Comparison of volume transport of Atlantic water through the N-section based on the new altimetry, calculated with and without in situ observations. Each square represents average transport for one month that had in situ data. The red squares show transport based on only altimetry plotted against transport based on altimetry and PIES measurements for 19 months. Cyan squares show transport based on only altimetry plotted against transport based on altimetry and bottom temperature measurements at NE ($T_{NE}$) for 115 months. The diagonal line indicates equality. In the upper left corner are shown the correlation coefficients for the red squares and for the cyan
squares. **(b)** Monthly volume transport through the N-section based on the old altimetry (and in situ) plotted against transport based on the new altimetry (and in situ) for 1993–2020 with the correlation coefficient ($R$) shown.

When the isotherm depth based on situ observations is used for calculating monthly volume transport (abscissa in Fig. 13a) the result deviates from transport calculated with altimetry only (ordinate in figure). The deviations are not large, however, and the correlations are high. One reason for this is, no doubt, that the velocities typically are low at the depth of





the 4 °C isotherm. Therefore, the transport is not very sensitive to the exact depth of the isotherm. A similar argument may be used for the northern boundary.

With the new altimetry data set, SLA values have been reprocessed and modified for the whole altimetry period. This necessitated the modification of existing algorithms and re-calculation of volume transport throughout the period. As documented in Fig. 13b, the changes in transport due to the altimetry reprocessing are, however, small.

**7 Discussion and conclusions**

In this section, the results from the four preceding sections are discussed. The main conclusions are enhanced in ***bold italic***.

**7.1 Comparison of in situ observations and old as well as new altimetry data**

In Sect. 3 (Table 3 and Table 4) and Sect. 6 (Table 10 and Table 11), several relationships between altimetry and in situ observations on the N-section were explored, using both the old and the new SLA data sets. ***Altogether, 20 correlation coefficients were calculated using both the old and the new data sets. In every single case, the correlation coefficient increased when going from the old to the new SLA data set***.

The primary relationship to investigate is between surface velocity and sea level slope. When this relationship is tested by comparing extrapolated ADCP surface velocities with SLA-differences between appropriate altimetry grid points, the correlation coefficients vary widely, even with the new altimetry (Table 3). Partly, this may be because the ADCP-derived velocities are not horizontal averages in contrast to altimetry-derived velocities. For two of the intervals ($A_3$–$A_4$ and $A_4$–$A_5$), we have sufficient data from four ADCP sites that may be combined to generate eastward surface velocities that approximate horizontal averages, Eq. (4) and Eq. (6). For monthly (28-day) averaged data, the correlation coefficients for these two intervals are 0.86 and 0.92, respectively, when using the new altimetry (Table 4).

A high correlation between two time series means that they are linearly related, but the coefficients may not necessarily be according to theory. For the intervals $A_3$–$A_4$ and $A_4$–$A_5$, this can again be tested by regression analysis. When this was done with the old altimetry, the regression coefficient was too high for both intervals by 36 % and the theoretical value was outside the 95 % confidence limits of the regression coefficient. With the new SLA data, in contrast, the agreement was almost exact, and the theoretical value was within the (narrow) confidence limits of the regression coefficient (Table 4).

Remarkably, the regression coefficients for the two intervals in Table 4 were almost identical. With the new SLA data, both regression coefficients were ≈6 % higher than the theoretical value. Whether this indicates that there still is a small bias in the new SLA data cannot be determined from these results since the theoretical coefficient was within the confidence limits for both intervals.

These good correspondences mean that both the ADCP extrapolation method (Fig. 3) and the method for generating horizontally averaged ADCP-velocity by Eq. (4) and Eq. (6) must be fairly accurate. Both of these methods will, however, introduce uncertainties into the ADCP-based surface velocities, which may be expected to degrade both correlation and





regression coefficients. The good correlation and regression coefficients in Table 4 would therefore likely have been even better, if we had more accurate in situ observations, with which to compare the SLA-data. ***For this ocean region, at least, we may conclude that the reprocessing involved in producing the new SLA data set has significantly improved its quality and***
***surface velocity anomalies calculated from the new SLA data appear highly accurate on monthly time scales.***

Since the SLA data represent sea level anomalies, they can only be used to calculate velocity anomalies. To determine absolute velocities, more information is needed. In theory, this can be provided by the MDT, but that requires that the MDT (including the geoid) is accurately known. Altimetric offsets, $U_k^0$, for the intervals $A_3$–$A_4$ and $A_4$–$A_5$, based on the MDT, disagree with the estimates based on ADCP data. The disagreement is smaller with the new MDT than with the old, but the
MDT-based values are still far outside the confidence limits of the ADCP-based values (Fig. 11).

This disagreement might be due to errors in the ADCP-based values, especially caused by the method used for extrapolating ADCP velocities to the surface (Fig. 3). If that were the explanation, however, it is difficult to understand how the regression coefficients can be so close to their theoretical values as argued above. According to Table 4, the regression coefficients for the intervals $A_3$–$A_4$ and $A_4$–$A_5$, are only ≈6 % higher than the theoretical values (and within confidence
limits). In contrast, the altimetric offsets for these two intervals based on the MDT are ≈25 % smaller than the ADCP-based values (Fig. 11).

On larger scales (across several grid points), the disagreement between MDT-based and ADCP-based altimetric offsets is not as large (Fig. 11). For the whole interval between $A_3$ and $A_8$, the average offset based on the new MDT is 11.6 cm s$^{-1}$, whereas the ADCP-based value is 12.1 cm s$^{-1}$. The small difference between these two values might lead one to think that
the transport through the whole section is not sensitive to the method used for estimating the altimetric offsets. The Atlantic layer is, however, deepest in the region (between $A_3$ and $A_5$) where the disagreement is large. Using $U_k^0$ values based on the new MDT, the average volume transport of Atlantic water through the N-section (1993–2018) would have been 3.0 Sv instead of the 3.8 Sv that are obtained by using $U_k^0$ values based on in situ measurements. If we used the old MDT, the average transport would have been even lower: 2.8 Sv.

It is well known that determination of the MDT is especially difficult in areas where strong currents are located over steep topography (Rio et al., 2011). For the flow through the N-section, ***our results indicate that the new MDT (Mulet et al., 2021) may be fairly accurate on spatial scales exceeding 100 km, but too smooth to accurately represent the strong flow over the slope north of the Faroes*** (Fig. 4).

## 7.2 The large-scale flow pattern of the IF-inflow

Combining the results from various sources (Sect. 4 and Sect. 5), it appears that the inflow across the IFR may be seen in terms of two separate branches: an "Icelandic branch" and a "Faroese branch" (Fig. 14). According to the new MDT, the Icelandic branch is a broad flow between altimetry points $I_1$ and $I_3$ with the average cross-ridge surface velocity being the same, 10 cm s$^{-1}$, for the $I_1$–$I_2$ interval and the $I_2$–$I_3$ interval. This is inconsistent with the high average surface velocity measured by the ADCP at site IW (Fig. 4), with the narrow drifter path (Fig. 5b), and with the analysis in Sect. 5.3.




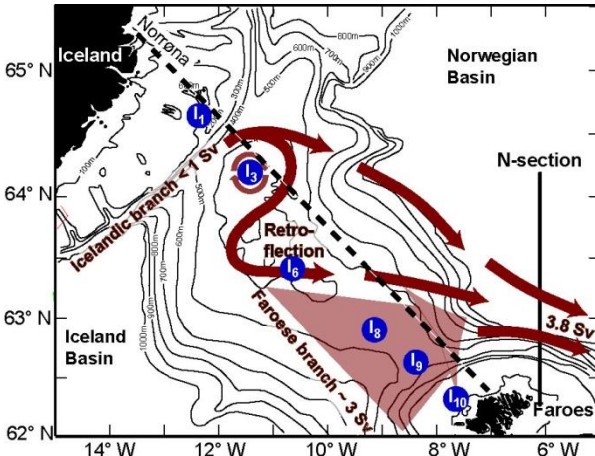

**Figure 14.** Schematic illustration of Atlantic water flow across the IFR with indications of average volume transport. Six of the altimetry grid points on the IFR are indicated. The dashed line is the typical track of the "Norröna" ferry.

Keeping in mind that small-scale variations of the MDT are questionable (Fig. 11), we choose to ignore the information from the MDT for this purpose. Instead, we find that by far most of the Icelandic branch passes through altimetry interval $I_1$–$I_2$ above the Western Valley (Fig. 8) as a narrow ($\approx$ 12 km), high-speed ($>$ 20 cm s$^{-1}$) current, topographically locked over the Icelandic slope close to the location of ADCP site IW with some of the flow leaking into interval $I_2$–$I_3$.

From Table 7, the average volume transport of the Icelandic branch is only 0.7 Sv, i.e., around one fifth of the total
inflow across the ridge. Even though this branch has by far the highest surface velocities, it is narrow (small $L_{Eq}$) and shallow (small $D_{Eq}$) compared to the flows comprising the Faroese branch.

An average transport value for the Icelandic branch below 1 Sv is less than suggested by the modelling study of Logeman et al. (2013), which had the "South Icelandic Current" crossing the ridge south of Iceland with an average (1992–2006) volume transport of 1.7 Sv. In their model, this flow supplies most of the transport of the Faroe Current, which they
estimate at 2.1 Sv, in clear disagreement with our results. Perkins et al. (1998) found an even higher Icelandic branch transport around 3.5 Sv, based on dynamic calculations on an unspecified set of CTD cruises. Rossby et al. (2018), on the other hand, found less than 0.5 Sv (estimated from their Fig. 4) of inflow in this region from vessel-mounted ADCP data along the track of the "Norröna" ferry between Iceland and the Faroes (dashed line in Fig. 14). Their data are from summer, only, when the Icelandic branch has a minimum (Fig. 9b), so their results are quite consistent with ours.

The low value for the Icelandic branch transport in Table 7 may also explain the previously mentioned (Sect. 1) controversy between Orvik and Niiler (2002) and Rossby et al. (2009). Orvik and Niiler (2002) focused on surface drifters with current speed $>$ 30 cm s$^{-1}$. This criterion will pick out the high-speed pathway over the Icelandic slope, but will not necessarily reflect volume transport, which should be better represented by the Rossby et al. (2009) study.





Shortly after passing ADCP site IW, the Icelandic branch appears to lose the topographical steering of the Icelandic

slope (Fig. 5b) and turns in a southeast-ward direction. According to the MDT, Some of this water continues in this direction, roughly following bottom contours, but some of it turns south- and westwards in a retroflection over central parts of the ridge and partly re-circulates over the northern part (Fig. 4). This will prolong the contact between the Atlantic water and the overflow water below it (Sect. 7.6), which may contribute to the strong cooling and freshening of the IF-inflow induced by crossing the IFR (Larsen et al., 2012).

The recirculation is also likely to affect biological processes in the region. On the IFR, the centre of the recirculation is located over the northernmost part of a bank, which was sufficiently interesting to German fishermen to be named the "Rosengarten Bank" (Fig. 14). The region is characterized by high surface chlorophyll concentrations in summer (e.g., Pacariz et al., 2016) and is known to be a mating area for deepwater redfish (*Sebastes mentella*, Melnikov and Popov, 2009).

Since small-scale variations of the MDT should be treated with caution (Sect 7.1), corroborating evidence for the

retroflection and recirculation would be advantageous. In "The Norwegian Sea" by Helland Hansen and Nansen (1909), there is an indication of a retroflection (their Fig. 32 and Fig. 39), but the surface circulation in the review by Meincke (1983) does not show this. Retroflection is indicated in the surface flow map by Beaird et al. (2016) and also by Rossby et al. (2018), but not as pronounced as indicated by the new MDT (Fig. 4). Our data set does not include any ADCPs in the region where the MDT indicates retroflection of water from the Icelandic branch back onto the IFR, but the drifters clearly

demonstrate that this process does occur, as exemplified by Fig. 5b. They also demonstrate that water does re-circulate over the northernmost bank, as indicated by the new MDT.

From Fig. 9b, the cross-ridge surface velocity through altimetry interval $I_3$–$I_6$ in the "outflow" region has a similar seasonal variation as the velocity through $I_1$–$I_3$, only oppositely directed. More generally, the retroflection and especially the recirculation increase with increasing strength of the Icelandic branch (Fig. 10). Since we lack reliable estimates of the

average flow between $I_3$ and $I_6$, it remains an open question whether the retroflection and recirculation typically are suspended during mid-summer or not.

The Faroese branch is not a narrow, topographically locked, jet, similar to the Icelandic branch. Rather, it covers a wide area over the southern part of the IFR, as indicated by the broad arrow in Fig. 14. From Table 7, the 190 km wide area between $I_6$ and $I_{10}$ has inflow with a total average volume transport of 3.3 Sv. According to Table 7, half of this flow enters

between $I_8$ and $I_9$, close to ADCP site IB, on average, but the location of crossing seems to vary. This is indicated by the negative correlation between the velocity at IB and the SLA-difference across $I_9$–$I_{10}$ in Table 5, as well as between velocity at IE and SLA-difference across $I_8$–$I_9$. These significantly negative correlations indicate that when the flow is strong between $I_8$ and $I_9$, it is weak both south of and north of this interval.

One way to interpret these correlations is for the Faroese branch to be a wide flow that is relatively stable in transport,

but meanders north and south between $I_6$ and $I_{10}$. Alternatively, the flow may be split into sub-branches, constrained by bottom topography, with variable strength of each sub-branch, but relatively stable total flow. This latter picture would be



consistent with the results from the Norröna ferry (Rossby et al., 2018), which show the average flow across the southern part of the IFR separated into three sub-branches (their Fig. 4).

### 7.3 Quality assessment of the altimetry-based IF-inflow monitoring system

When monitoring of the IF-inflow was initiated in the mid-1990s, the N-section was chosen partly because it had already been occupied by regular CTD cruises and partly because it crosses the flow after it has become much narrower. Monitoring on the N-section, after the modifications occurring over the IFR, has the added benefit that the transports and water mass properties are more representative of the heat and salt input to the Arctic Mediterranean and better indicators for regional components of the AMOC for climate assessments.

Calculation of volume transport for each month involves three main steps. The first step is to determine the average eastward surface velocity for each altimetry interval along the N-section for the month, Eq. (2). From the discussion in Sect. 7.1, this may be done with a high accuracy. Secondly, the deep boundary of the Atlantic water along the section is determined for each month from the altimetry data, Eq. (A3), as well as its northern limit, Eq. (A5) and Eq. (A6). As detailed in Sect. 6.3, this may again be achieved with a high accuracy. It is remarkable that the algorithms for calculating Atlantic

water depth are much more accurate for monthly averages (Table 11) than for the snapshot CTD observations, from which they were developed (Table 10). The final step is the vertical integration of eastward velocity from the surface down to the boundary, Eq. (A1). This is achieved by assuming proportionality between surface deeper velocities, Eq. (A2). From the ADCP data, this is a good approximation (Table 8). To some extent, this may be because of the proportionality assumed in the extrapolation method, Eq. (1) and Fig. 3, but the good correspondence between extrapolated ADCP velocity and SLA-

differences, demonstrated in Sect. 7.1, validates the method.

Although in situ information on the extent of the Atlantic layer can increase the accuracy of the volume transport, it thus appears that monthly averaged volume transport without in situ information is fairly accurate, as well. This is confirmed in Fig. 13a, where volume transport was calculated with and without in situ observations for months with these observations. Certainly, none of the squares in Fig. 13a represents in situ observation of isotherm depth at all stations along the section at

the same time, but together they cover the area with most of the transport.

*Summarizing, we conclude that the established system, based on SLA data, is able to generate accurate time series of monthly averaged volume transport for the whole altimetry period. With existing technology, a system based only on in situ observations would have to be prohibitively comprehensive and resource-demanding for it to perform better than the chosen system based on satellite altimetry. The accuracy obtained from the altimetry-based system is, however, only*

*possible because of the large set of previously acquired in situ observations that have been used to calibrate the altimetry data and develop the algorithms necessary for transport calculation. Also, in situ observations are still necessary for monitoring transport of heat, salt, and other substances.*

In addition to volume transport, the monitoring system also produces monthly estimates of the heat carried by the IF-inflow. For an individual inflow branch, the heat that it will deliver to a region like the Arctic Mediterranean is not well



defined, since it depends on the average temperature of the water when it leaves the region again; not all of it necessarily at the same time or the same location. Most of the Atlantic inflow will, however, later be converted into overflow water, which leaves with average temperatures close to 0 °C (Østerhus, et al., 2019). Using this value as a fixed reference temperature in heat transport calculation should therefore yield values close to the actual heat released in the region, as verified by the more rigorous analysis presented by Tsubouchi et al. (2021). We shall, however, use the term *"relative heat transport"* for this time series to emphasize that it is relative to 0 °C.

Algorithms for calculating the relative heat transport are listed in Eq. (A7) and Eq. (A8). They include altimetry data and are affected by the switch from the old to the new SLA data, but only through parameters included in calculation of volume transport, which have been previously discussed.

### 7.4 Time series of volume and relative heat transport of the IF-inflow 1993 – 2021

Monthly and annually averaged values for volume and relative heat transport are illustrated by the curves in Fig. 15a for the period January 1993 to December 2021. As discussed by Hansen et al. (2015), the method may introduce a systematic bias, leading to an uncertainty of ±0.5 Sv for the average volume transport. From the discussion in Sect. 6.1 and Sect. 7.3, there is no indication that this uncertainty has been underestimated and relative variations ought to be considerably more accurate. The time series illustrated by the curves in Fig. 15a, therefore, ought to give a fair representation of the variations of the IF-inflow, as it passes through the N-section.

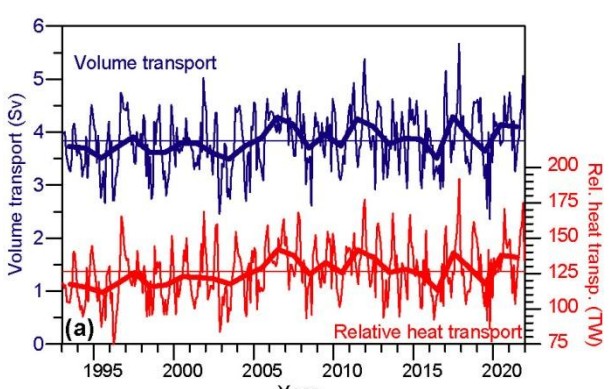
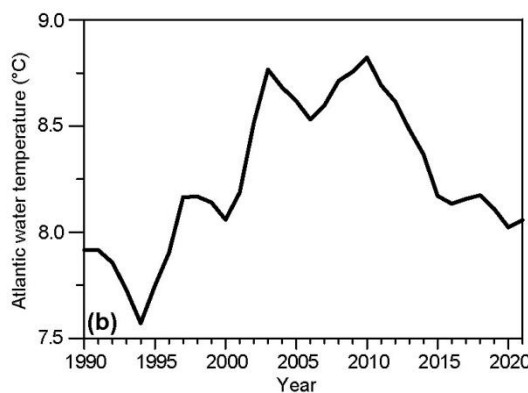

**Figure 15. (a)** Monthly (thin lines) and annually (thick lines) averaged volume (blue) and relative heat (red) transports. Thin horizontal lines show average values. The heat transport is relative to a temperature of 0 °C. **(b)** Three-year running mean Atlantic water temperature on the N-section.

Averages of the two transport time series over the whole 1993–2021 period (Table 12) are almost identical to the estimates by Hansen et al. (2015) for the 1993–2013 period and the seasonal variations are similar, as well (their Table 3). Both volume and relative heat transport have seasonal variations that are close to sinusoidal with maxima towards the end of the year and seasonal amplitudes between 11 % (volume transport) and 13 % (relative heat transport) of the average values (Table 12).





**Table 12.** Characteristics of the two time series of IF-inflow transport through the N-section. Uncertainty estimates for the average values are copied from Hansen et al. (2015). The seasonal variation is characterized by three parameters: "$R_{Seas}$" is the correlation coefficient with a sinusoidal, "$A_{Seas}$" is its amplitude, and "Max" the month of maximum transport. The parameters were determined by linear regression of high-passed (by subtracting 13-month running mean) transport values with a sinusoidal where the month of maximum was varied to give maximum correlation. The trends are listed with 95 % confidence intervals. "Change" indicates the relative (to average) change through the 1993–2021 period. 1 TW $= 10^{12}$ W.

| Time series | Average | $R_{Seas}$ | $A_{Seas}$ | Max. | Trend | Change |
|---|---|---|---|---|---|---|
| Volume transp.: | (3.8±0.5) Sv | 0.60 | 0.43 Sv | Dec. | (0.012±0.010) Sv yr$^{-1}$ | +9 % |
| Rel. heat transp.: | (126±15) TW | 0.64 | 16 TW | Nov. | (0.57±0.35) TW yr$^{-1}$ | +13 % |

Alternative transport values for the IF-inflow have been reported by Rossby et al. (2018), mainly based on data acquired during summer on the "Norröna" ferry (Fig. 14). Their summer-averaged volume transport, 4.8±0.7 Sv, is higher than our annual average, but the values overlap within uncertainty limits. Also, their value appears to include some transport over the Faroe shelf, which we consider recirculation around the Faroes and exclude (Hansen et al., 2015). This should reduce the discrepancy further. The average relative heat transport reported by Rossby et al. (2018) is also higher than ours, but again within the combined uncertainty estimates and not defined in quite the same way.

Considering potential interactions between the IF-inflow and the Arctic waters over and east of the IFR, there is no reason to expect that the total volume transport of the IF-inflow over the ridge should equal the transport through the N-section exactly. Nevertheless, the close correspondence between the sum in Table 7 (4.0 Sv) and the average N-section transport (3.8 Sv) is comforting.

**Table 13.** Correlation coefficient between annually averaged volume transport through the N-section and SLA-difference between two altimetry points on the IFR (latter minus former).

| $I_1$-$I_{10}$ | $I_1$-$I_3$ | $I_3$-$I_6$ | $I_4$-$I_6$ |
|---|---|---|---|
| 0.31* | 0.41* | -0.39* | -0.45** |

From Fig. 9b, the flows across different parts of the IFR have different seasonal variations. The sea level slope across the whole ridge ($I_1$–$I_{10}$) and the volume transport through the N-section also differ in seasonality (Fig. 9b and Table 12). It therefore seems futile to correlate monthly averages of these time series, but annually averaged volume transport through the N-section is significantly correlated with the SLA-difference across the entire width of the IFR (Table 13). Remarkably, the correlation with interval ($I_1$–$I_3$) is considerably higher than with ($I_1$–$I_{10}$). By Eq. (8), this means that the volume transport of the Icelandic branch is significantly correlated with the transport through the N-section on inter-annual time scales. The standard deviation of the Icelandic branch transport (0.46 Sv) is also almost as high as the standard deviation of the N-section transport (0.55 Sv). On inter-annual time scales, the Icelandic branch, thus, contributes considerably to the variability of the N-section transport, even though it contributes little to the average transport (Table 7 and Fig. 14).



## 7.5 Long-term variations of the IF-inflow

Over the whole 1993–2021 period, both volume transport and relative heat transport through the N-section had statistically significant increasing trends. The increases are not equally distributed over the period and may alternatively be seen as a step
change around 2001 (Tsubouchi et al., 2021) when volume transport increased from 3.7 Sv to 3.9 Sv and relative heat transport increased from 118 TW to 129 TW. From the last column in Table 12, the percentage change over the whole period is higher for the relative heat transport than for the volume transport, illustrating that the increase in relative heat transport derives from increased Atlantic water temperature as well as from increased volume transport, Eq. (A7) and Eq. (A8). Figure 15b illustrates the long-term variation of the Atlantic water temperature on the N-section, which is defined as the average
temperature of the layer between 100 m and 150 m depth at standard station N03 (Hansen et al., 2015). Although statistically significant, the trends in Table 12 are not much higher than the 95 % confidence limits, but it still seems fair to conclude that ***neither the volume transport nor the relative heat transport weakened during the monitoring period 1993–2021.***

According to Østerhus et al. (2019), 70 % of the total Atlantic inflow to the Arctic Mediterranean is converted to overflow and the volume transport of the IF-inflow contributes almost one half. The overflow is the main source of high-
density water to the deep limb of the AMOC, although its volume transport is strongly enhanced south of the GSR by ventilation and entrainment (Dickson and Brown, 1994; Sarafanov, 2012; Lozier et al., 2019; Koman et al., 2022).

Long-term variations of the total Atlantic inflow are therefore intimately linked to the overflow and to the AMOC. The AMOC is projected to weaken during the 21$^{st}$ century (Arias et al., 2021) and reports have claimed that it has already weakened at 26° N during our observational period (Smeed et al., 2014; 2018). Updated estimates from 26° N report
recovery of the AMOC, but they still report a period around 2010 with a weakened AMOC at 26° N (Moat et al., 2020; Worthington et al., 2021). This weakening was especially pronounced for the Lower North Atlantic Deep Water component, fed by the overflows (Smeed et al., 2014).

One might therefore expect to see a similar period of weakened overflow and Atlantic inflow, but the IF-inflow has no indication of this (Fig. 15a). Although the main inflow branch to the Arctic Mediterranean, the IF-inflow is, of course, only
one part of the total inflow, but Østerhus et al. (2019) found no extended period with pronounced weakening in the observed total Atlantic inflow between 2000 and 2014. Consistent with that, they also reported a relatively stable volume transport of the two main overflow branches during this period (their Fig. 10). Updated time series of observed transports combined with results from several reanalysis products show no indication of a multi-year negative anomaly in total Atlantic inflow around 2010 (Mayer et al., accepted). Thus, the relative stability of the IF-inflow during the whole period from 1993 to the end of
2021 is consistent with other observational evidence on the flows across the Greenland-Scotland Ridge.

## 7.6 Overflow-inflow coupling over the IFR and its implications

The negative correlations in the last two columns of Table 13 imply that years with strong inflow through the N-section have stronger than normal outflow (or weaker inflow) in the retroflection region over the southern Rosengarten Bank (Fig. 4)



between $I_3$ and $I_6$. This rather surprising result may help shed light on a problem that is linked to the coupling between IF-
inflow and IFR-overflow. As stated in Sect. 1, this study does not treat the IFR-overflow per se, but the overflow-inflow
coupling may affect the predictability of the IF-inflow, which motivates a closer look at the IFR-overflow.

Although identified already in the 19[th] century (Knudsen, 1898), many aspects of the IFR-overflow are still unknown.
An ADCP deployed over the Icelandic slope downstream of the IFR from 2005 to 2007 (blue circle labelled "O" in Fig. 16)
showed a south-westward current with a core 50 m above the bottom in a layer with overflow water properties and with
average speed exceeding 50 cm s$^{-1}$. Over more than two years, there was not one week with core-speed less than 30 cm s$^{-1}$
(Voet, 2010; Olsen et al., 2016). Also, Perkins et al. (1998) had previously observed this strong flow over more than six
months in 1991–1992. Comparison with measurements downstream of the Faroe Bank Channel (Geyer et al., 2006; Darelius
et al., 2011; 2015; Ullgren et al., 2016) indicates that FBC-overflow is not the source of this flow and Beaird et al. (2013)
argued that it had to derive from overflow across the northern part of the IFR.


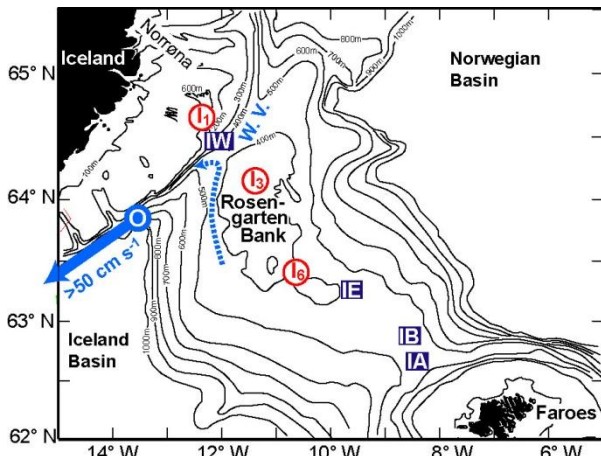

**Figure 16.** The thick blue arrow indicates a bottom-intensified current with average speed more than 50 cm s$^{-1}$ at the core measured by an
ADCP (blue circle labelled "O") for more than two years 2005–2007 (Olsen et al., 2016). The dashed blue arrow is the flow suggested to
feed this current by Perkins et al. (1998). "W. V." is the Western Valley. Three altimetry grid points and four ADCP sites on the IFR are
indicated.

This argues that there is persistent overflow across the IFR. All of our ADCP moorings on the IFR have average bottom
temperatures below 3 °C and all of them have frequent near-bottom velocities in the overflow direction (Table 14), but none
of them show persistent overflow (Østerhus et al., 2008; Hansen et al., 2018) that would explain a persistency downstream as
seen in 2005-2007. It has been suggested that the lack of persistent overflow through the Western Valley is due to blocking
by the overlying IF-inflow (Hansen et al., 2018) and this mechanism may perhaps also explain the lack of persistent
overflow at sites IA, IB, and IE, as well. ***At all of these sites, there are significant positive correlations between the cross-***



*ridge velocities near the surface and near the bottom (Table 14), supporting the hypothesis of a dynamical link between*
*the inflow and the overflow directly below.*


**Table 14.** Overflow-relevant characteristics at the four ADCP sites on the IFR. "$T_{Bottom}$" is the average temperature at the ADCP, close to
the bottom. "Outflow" is the percentage of days with negative cross-ridge velocity at the deepest bin. "Correlation" is the correlation
coefficient between the cross-ridge velocity at the deepest bin and the uppermost bin (Top depth, Sect. 2.2) for data averaged over one day
and seven days, respectively.

| Site | $T_{Bottom}$ | Outflow | Correlation | |
| | | | 1 day | 7 days |
|---|---|---|---|---|
| IA | 2.3 °C | 47 % | 0.57*** | 0.79*** |
| IB | 1.8 °C | 37 % | 0.39*** | 0.39** |
| IE | 0.6 °C | 50 % | 0.40*** | 0.35* |
| IW | 0.9 °C | 69 % | 0.30*** | 0.39* |


These arguments indicate that the primary origin of the persistent overflow observed downstream of the IFR may be
over the southern part of the Rosengarten Bank (Fig. 16) where the average surface flow is towards the Iceland Basin
according to the MDT (Fig. 4) even though this region is relatively shallow. If that is the case, it provides support for an
hypothesis put forward by Olsen et al. (2016) to explain the discrepancy between observed and simulated volume transport
through the N-section.

As stated in Sect. 1, numerical ocean models (particularly in coarse configurations for climate sensitivity studies) have
not been very successful in replicating the observed transport variations of the IF-inflow, which also will limit its
predictability in coupled climate models. Olsen et al. (2016) suggested that the reason might be the inability of models with
even relatively high resolution to simulate IFR-overflow adequately. The simulated IF-inflow might therefore in reality be
the net inflow (Atlantic water in minus overflow out). If the Atlantic inflow and the overflow were to be positively
correlated, the two contributions to net inflow would partially cancel one another. The deployment of the ADCP at site IW
was partly motivated by a hope to verify this hypothesis, but the results seemed rather to invalidate it since the inflow and
the weak overflow through the Western Valley were found to be negatively correlated (Hansen et al., 2018).

With the new results presented here, especially the correlations in Table 13, the question is re-opened and a mechanism
may be proposed: In years with high inflow through the N-section, the inflow through the Western Valley will also tend to
be high (positive correlation), but so will the south-westward retroflection over the southern Rosengarten Bank (negative
correlations). The strengthened surface outflow may be expected to stimulate the overflow. ***This may explain the positive***
***correlation between Atlantic inflow and overflow over the IFR needed to support the Olsen et al. (2016) hypothesis. This***
***is, however, still an hypothesis and more observations and high-resolution modelling will be needed for more certain***
***validation.***



## Appendix A: The methodology for calculating monthly transport values

Following the strategy established in Hansen et al. (2015), calculation of Atlantic water volume transport, *Q(t)*, in the Faroe Current is based on horizontal and vertical integration along the altimetry line $A_2$ to $A_8$ extending northwards on the N-section along the 6.125° W longitude (Fig. 2):


$$Q(t) = \sum_{k=2}^{7} \sum_{z=1m}^{500m} U_k(z,t) \cdot W_k(z,t) \tag{A1}$$

where $U_k(z,t)$ is the eastward velocity at depth *z* and time *t* horizontally averaged within altimetry interval *k*, which spans $A_k$–$A_{k+1}$ for *k* ranging from 2 to 7. $W_k(z,t)$ is the width of altimetry interval *k* at depth *z* and time *t*. When the whole interval is

within the Atlantic water domain, the width is equal to the distance between the two altimetry points at each end of it (except for the southernmost interval, which starts in the middle to exclude the Faroe shelf). At greater depth, the width starts to decrease when the bottom or the deep boundary of the Atlantic layer is reached, and the width falls to zero at depths where the whole interval is below the deep Atlantic water boundary or the bottom. Similarly, the width is reduced when the interval extends north of the northern boundary of Atlantic water.

To determine *Q*(t) for any given month, we therefore need the average velocity field for that month $U_k(z,t)$ and the extent of Atlantic water on the section, from which $W_k(z,t)$ is easily derived. For the velocity field, it was suggested in Hansen et al. (2019) that the eastward velocity at depth *z* to a good approximation is proportional to the eastward surface velocity for the same month:

$$U_k(z,t) = \Phi_{k,m}(z) \cdot U_k(0,t) \tag{A2}$$

where the proportionality factor, $\Phi_{k,m}(z)$, for each altimetry interval, *k*, and month, *m*, was determined in Hansen et al. (2019). $U_k(0,t)$ is the eastward surface (*z* = 0) velocity between grid points $A_k$ and $A_{k+1}$, horizontally averaged over the interval and is determined by Eq. (2).

Determination of monthly values for $W_k(z,t)$ requires determination of the Atlantic water extent on the section; both vertically and horizontally. The Atlantic water extent on the N-section is defined by its depth at each of the standard stations from N02 to N10 and by the latitude of its northern boundary. South of standard station N04, the Atlantic water extends all the way to the bottom according to observations. From N04 to the northern boundary, the downward extent is defined by the 4 °C isotherm, slightly adjusted for variations of Atlantic water core temperature by reducing the depth by 15 m for every

degree that the core is warmer than 8 °C (Hansen et al., 2020). The depth of this isotherm has consistent long-term and seasonal variations plus a short-term variation, which is expressed in terms of altimetry data. The algorithm for determining the isotherm depth at each standard station was determined from the CTD data for N04 to N10 by multiple regression analysis (Hansen et al., 2020):





$D_j(t) = D_j^0 + \gamma_j \cdot t + a_{\mathrm{TA,j}} \cdot [T_A(t) - \langle T_A \rangle] + A_j \cdot \cos\left[2\pi \cdot \left(t - \frac{Day_j}{365}\right)\right] + a_{\mathrm{h,j}} \cdot h_j(t) + a_{\mathrm{x,j}} \cdot Pc_1(t)$      (A3)

where $D_j(t)$ is the depth of the 4 °C-isotherm at standard station $N_j$ at time $t$ expressed in years, $T_A(t)$ is the 3-year running mean of the de-seasoned average temperature 101–150 m at N03, $\langle T_A \rangle$ is the average of $T_A(t)$ between 1993 and 2017 (8.336 °C), $h_j(t)$ is the sea level anomaly at the location of the station, derived by linear interpolation between neighbouring

altimetry grid points, and $Pc_1(t)$ is the principal component of the first EOF (Empirical Orthogonal Function) mode of the $H_k(t)$ values. The EOF analysis is documented in Hansen et al. (2020) (their Sect. 5.2) where this parameter was termed *PcAH-1(t)* and where it is seen that this principal component explains 88 % of the variance of sea level height. The coefficients in Eq. (A3) were originally determined from the old altimetry data (Hansen et al., 2020). Updated coefficients to use with the new data set are listed in Table A1.


**Table A1.** Coefficients to use with Eq. (A3) to estimate 4 °C-isotherm depth at stations N04 to N10 when using the new altimetry data.

| Coeff.: | $D_j^0$ | $\gamma_j$ | $a_{TA,j}$ | $A_j$ | $Day_j$ | $a_{h,j}$ | $a_{x,j}$ |
|---|---|---|---|---|---|---|---|
| Unit: | m | m yr$^{-1}$ | m °C$^{-1}$ | m | | | m |
| N04: | 368 | 1.97 | 0.0 | 25 | 298 | 1561 | -61.29 |
| N05: | 261 | 2.33 | 30.6 | 32 | 294 | 2739 | -115.92 |
| N06: | 205 | 3.20 | 44.0 | 45 | 283 | 2167 | -106.16 |
| N07: | 162 | 3.16 | 45.1 | 65 | 262 | 1902 | -88.02 |
| N08: | 115 | 3.12 | 30.4 | 56 | 269 | 1946 | -78.16 |
| N09: | 59 | 3.00 | 0.0 | 48 | 262 | 917 | 0.00 |
| N10: | 48 | 1.21 | 0.0 | 48 | 270 | 493 | 0.00 |

For most of the standard stations, Eq. (A3) explains a considerable fraction of the variance, but not for station N04. In periods when daily averaged bottom temperature at site NE, $T_{NE}(t)$, is available (e.g., from an ADCP temperature sensor),

determination of the 4 °C-isotherm depth at station N04 is improved by the equation:

$D_4(t) = d_{0,4} + \gamma_4 \cdot t + A_4 \cdot \cos\left[2\pi \cdot \left(t - \frac{Day_4}{365}\right)\right] + a_{\mathrm{NE}} \cdot T_{NE}(t) + b_{\mathrm{NE}} \cdot [H_3(t) - H_4(t)]$      (A4)

with the coefficients listed in Table A2.


**Table A2.** Coefficients to use with Eq. (A4) to estimate 4 °C-isotherm depth at stations N04 when using the new altimetry data together with bottom temperature at NE.

| Coeff.: | $d_{0,4}$ | $\gamma_4$ | $A_4$ | $Day_4$ | $a_{\mathrm{NE}}$ | $b_{\mathrm{NE}}$ |
|---|---|---|---|---|---|---|
| Unit: | m | m yr$^{-1}$ | m | | m °C$^{-1}$ | |
| | 264 | 1.97 | 25 | 298 | 31.2 | -789 |



The northern boundary of Atlantic water on the section, $B_j(t)$, is a real number between 4 and 10, which is in units of

standard stations (e.g., $B_j(t) = 7.5$ means that the boundary is midway between N07 and N08). It is defined to be where the

"normalized maximum salinity", $S_j^*(t)$, falls below 35.075 (Hansen et al., 2020). To a good approximation:

$$S_j^*(t) \cong \langle S_j^* \rangle + M_j^{S1} \cdot PcS_1(t) \qquad (A5)$$

where $\langle S_j^* \rangle$ is the average, $M_j^{S1}$ is the spatial variation of the first EOF mode of the normalized maximum salinity, and $PcS_1(t)$

is the associated principal component. As shown in Hansen et al. (2020), this principal component may be well estimated

from the altimetry data with an expression, which with the new altimetry data has the form:

$$PcS_1(t) \cong [a_S \cdot H_6(t) + 1.035 \cdot Pc_1(t) + 0.905] \qquad (A6)$$


where $a_S = -21.1$ m$^{-1}$ and $Pc_1(t)$ again is the principal component of the first EOF mode of the $H_k(t)$ values. The values for

$\langle S_j^* \rangle$ and for $M_j^{S1}$ are listed in Table A3.

**Table A3.** Values of the average and of the first EOF mode for the normalized salinity maximum along the section when using the new
altimetry data.

|                      | N04     | N05     | N06     | N07     | N08     | N09     | N10     |
|----------------------|---------|---------|---------|---------|---------|---------|---------|
| $\langle S_j^* \rangle$: | 35.249  | 35.217  | 35.192  | 35.174  | 35.114  | 35.030  | 34.998  |
| $M_j^{S1}$:          | -0.0128 | -0.0218 | -0.0437 | -0.0849 | -0.1018 | -0.0723 | -0.0277 |

Monthly averaged values for heat transport relative to 0 °C, $\Omega(t)$, are calculated by the expression:

$$\Omega(t) = \rho \cdot C \cdot \sum_{k=2}^{7} \sum_{z=1m}^{500m} U_k(z,t) \cdot T_k(z,t) \cdot W_k(z,t) \qquad (A7)$$


where $\rho \cdot C$ is the heat capacity per cubic meter while $T_k(z,t)$ is the temperature at depth $z$ and time $t$ horizontally averaged

across altimetry interval $k$. The values for $T_k(z,t)$ are derived by linear interpolation between the temperature, $T_j(z,t)$, at

standard stations. $T_j(z,t)$ is the temperature at depth $z$ on standard station $j$ for time $t$ (in years) and is found as:

$$T_j(z,t) = T_j^0(z) + A_j^T(z) \cdot \cos\left[2\pi \cdot \left(t - \frac{Day_j^T(z)}{365}\right)\right] + a_j^T(z) \cdot [T_A(t) - \langle T_A \rangle] + b_j^T(z) \cdot x_j(t) + c_j^T(z) \qquad (A8)$$

where: $x_j(t) = \begin{cases} Pc_1(t) & for\ j < 4 \\ D_j(t) & for\ j \geq 4 \end{cases}$



*Data availability*. Data from in situ observations used in the study are available on www.envofar.fo

*Author contributions*. KMHL and BH coordinated acquisition of in situ observations. BH wrote the manuscript with input from all authors.

*Competing interests*. The authors have no competing interests.

*Acknowledgements*. The in situ observations reported here have been supported by the Danish Ministry of Climate, Energy and Utilities through its climate support program to the Arctic, by "Jens Smeds Oceanografiske fond", Denmark, by The

National Centre for Climate Research (NCKF) hosted at the Danish Meteorological Institute, Denmark, and by the Blue Action project (EU Horizon 2020 grant agreement nr. 727852). This study has been conducted using E.U. Copernicus Marine Service Information; https://doi.org/10.48670/moi-00148.

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
