# Peer review of "The Iceland-Faroe warm-water flow towards the Arctic estimated from satellite altimetry and in situ observations"

_EGUsphere, 2023_

## Author Comment (AC1)

**Referee 1**

We thank the referee for a very constructive and helpful review. Below, we detail how each referee comment has been addressed and how the manuscript has been modified accordingly.

**Manuscript structure**

As suggested by both referees, we have tried to re-structure the manuscript to enable better flow. Five tables have been moved to a new Appendix B and Figure 12 has been moved to the Materials and methods section together with associated text. In addition, the Introduction (Sect. 1) has been organized into four sub-sections, mainly to put more emphasis on the definition of Atlantic water for transport calculation.

In the following, we address specific comments from Referee 1 where quoted text from the old or the new version of the manuscript is in *italic.*

**Organization:**

**Referee:** The results sections 5 and 6 (having multiple sub-sections) would benefit from a brief introductory paragraph for each of those sections (before 5.1 and 6.1).
**Reply:** Has been done

**Referee:** The results sections include a lot of detail on methods used (for example section 6.1 which refers back to section 3 and Table 3 a few times; section 6.2 following on from 6.1; parts of section 6.3 (for example Fig 12)). Would the paper flow better if some sections got moved into the Methods section? Maybe that will disturb the flow of the paper but maybe the authors could consider this.
**Reply:** Fig. 12 and associated text has been moved to Sect. 2.1. Together with five tables being moved to the new Appendix B, this should help streamlining the manuscript. Unfortunately, we have not been able to identify any additional re-structuring that could get a better flow.

**Referee:** Tables: a total of 14 tables are in the main part of the manuscript, as mentioned above, could these get critically evaluated to see if some could get moved to the methods section and/or an appendix (Tables 3, 4 include a lot of detail that it already suitably summarized in the text and could be included in an Appendix for example).
Reply: A new appendix was added: "Appendix B: Supplementary tables" and 5 tables moved from the main text to this appendix: Table 1, Table 3, Table 5, Table 10, and Table 14. The associated text was modified to de-emphasize these tables.

**Referee:** Finish the paper with one strong paragraph summarizing the main results in a final Conclusions section (7.7).
**Reply:** Has been done. The heading of Sect. 7 has been modified accordingly.

**Clarifications:**

**Referee:** Lines 145-146: What is the seasonal spread of the CTD cruises? Have you applied standard QC to the CTD data, if so, mention it.
**Reply:** The text: "*mainly in February, May, August–September, or November*" has been added. Also, the following sentence was added: "*Initially, an EG&G CTD was used but since 1996, this was replaced by a SeaBird 911+. Water samples were acquired for salinity calibration and all the data have been quality controlled.*"

**Referee:** Lines 169-174: What type of ADCP was used?
**Reply:** The following sentence has been added: "*Three different ADCP models from Teledyne RD Instruments have been used: 150 kHz Broadband, 75 kHz Broadband, and Long Ranger.*"

**Referee:** Line: 175: "either 10 m or 25 m" state what this depends on.
**Reply:** We have added the text: "*depending on bottom depth and ADCP model*"

**Referee:** Line 207: "necessary additional information" – be specific what this it.
**Reply:** The paragraph has been reorganized and it has been specified that it is the lack of regular CTD data that prevent a more reliable extrapolation to the surface.

**Referee:** Lines 362-363: the average surface velocity is likely underestimated by the extrapolation method – could you give an indication or estimate of how much this would be?
**Reply:** We do not have any evidence, on which to base an objective estimate and would prefer not to guess. Instead, we now write: "*the shape of the profile indicates that the average surface velocity is likely to be underestimated by the extrapolation method, although it is difficult to estimate by how much.*".

**Referee:** Lines 683-684: Here you use more decimals than in the rest of the paper, be consistent, 11 cm s$^{-1}$ and 12 cm s$^{-1}$ .
**Reply:** We used the extra decimal to indicate that the difference between the two numbers was small, but the sentence has been changed so that it now reads: "*For the whole interval between $A_3$ and $A_8$, the average offset based on the new MDT is only 4 % smaller than the ADCP-based value.*"

**Referee:** Sections 7.2 and 7.4 need highlights in bold italic.
**Reply:** Section 7.2 now has three highlighted sentences and Section 7.4 has two.

**Figures:**

**Referee:** It would be useful if the lat/lon range in the figures were consistent (except for 1a which requires a larger area for context) for ease of comparison (especially for Figs 5a and 5b).
**Reply:** Old Fig. 5a (new Fig. 6a), Fig. 5b (new Fig. 6b), Fig.14, and Fig. 16 now have the same map boundaries and the same basic map.

**Referee:** Please make the colour bars distinct for ease of interpretation (Figs 4, 10).
**Reply:** The colour bars of old Fig. 4 (new Fig. 5) and Fig. 10 (new Fig. 11) have been made distinct by introducing lines at the same intervals as the contour lines.

**Referee:** Are the figures compatible with colour-vision deficiencies? (using not just colours but dashed/dotted lines for example).
 **Reply:** Old Fig. 6 (new Fig. 7) and Fig. 9b (new Fig. 10b) have been modified by making some of the lines dashed.

**Referee:** Fig 3a: could this figure get rotated to have depth on the y-axis to match Fig 3b or is there a specific reason for this orientation?
**Reply:** Old Fig 3a (new Fig. 4a) has been rotated so that the depth axis is vertical as in Fig 3b.

**Referee:** Fig 6: Mark (by a horizontal line?) where the extrapolation starts on each of the four profiles.
**Reply:** The extrapolation start is now marked by a black circle on each profile on the new Fig. 7a.

**Referee:** Fig 7: make the four sub-plots consistent with regard to x-lim, number of bars for ease of comparison.
**Reply:** A new consistent figure (new Fig. 8) has been made.

**Technical corrections:**

**Referee:** Line 35: Define Greenland-Scotland-Ridge acronym (GSR) at first use and then use acronym throughout.
**Reply:** The Greenland-Scotland-Ridge is only referred to twice in addition to this, and towards the end of the manuscript. We have therefore instead replaced the one reference to "*GSR*" by the full name: "*Greenland-Scotland Ridge*".

**Referee:** Line 41: include abbreviation "IFF" here.
**Reply:** Has been done.

**Referee:** Line 43-44: check the grammar of that sentence.
**Reply:** The sentence has been split into two sentences and clarified.

**Referee:** Line 143: "Materials and Methods"
**Reply:** Has been corrected.

**Referee:** Line 146: "many CTD profiles" – be specific: "between 98 and 155 CTD profiles"
**Reply:** Has been specified.

**Referee:** Line 229: only use "MDT" (acronym already defined earlier)
**Reply:** Has been done.

**Referee:** Fig 2a: The label for A2 is hidden behind N01, reposition it so it is visible; label for A4 is missing in the figure
**Reply:** The figure has been modified to correct this.

**Referee:** Fig 5b: the blue track is hard to see, choose a different/more distinct colour/line style/width.
**Reply:** The track has been made thicker and the background has also been changed, which should enhance the contrast (new Fig. 6b).

**Referee:** Line 643: change to "Comparison of in situ observations with old and new altimetry data"
**Reply:** Has been changed

**Referee:** Line 725: "some"
**Reply:** Has been corrected.

---

## Author Comment (AC2)

**Referee 2**

We thank the referee for a very constructive and helpful review. Below, we detail how each referee comment has been addressed and how the manuscript has been modified accordingly.

**Manuscript structure**
As suggested by both referees, we have tried to re-structure the manuscript to enable better flow. Five tables have been moved to a new Appendix B and Figure 12 has been moved to the Materials and methods section together with associated text. In addition, the Introduction (Sect. 1) has been organized into four sub-sections, mainly to put more emphasis on the definition of Atlantic water for transport calculation.

In the following, we address specific comments from Referee 2 where quoted text from the old or the new version of the manuscript is in *italic.*

**Referee:** The manuscript is however very technical with several analysis and many correlations are presented. The manuscript will benefit from moving some of these analysis/tables to supplementary material or to an appendix. This will make it easier for the reader to follow the steps and analysis.
**Reply:** Five of the tables have been moved to a new Appendix B. Also, Fig. 12 and associated text has been moved to Sect. 2.1.

**Major comment**

**Referee:** The transport estimates are calculated by integration the velocities down to the depth where T=4 $^o$C that they define as the base of the Atlantic layer. However, the properties of the AW change with time (e.g., fig 15b), and the authors should give some estimates of the sensitivity of results if other depths are used as the lower limit. E.g., what will the transport estimates be when using T=3 $^o$C or T=5 $^o$C as the lower limit of the depth integration.
**Reply:** The discussion of Atlantic water in the Introduction has been enhanced to include the sensitivity and the effect of changing AW properties. In connection with this, the Introduction has been split into four subsections, for emphasis, with the discussion of Atlantic water as Sect. 1.3.

**Minor comments**

**Referee:** Is table 1 needed? Fig. 2. shows the bottom depth of the area/stations and it is already written that the CTD-stations are taken 3-4 times a year. If deleted, the sentence on lines 146-147 can also be deleted.
**Reply:** Table 1 has been moved to the new Appendix B.

**Referee:** Line 211-212: "… the extrapolation factor **may be** modified to account for these." Please, specify. Was this done?
**Reply:** "*may be modified*" has been changed to: "*was modified*"

**Referee:** Figure 6. It would be interesting to see the variance of the cross-ridge velocities for the four ADCPs, e.g., include standard deviation in the figure.
**Reply:** A graph showing standard deviation has been added (new Fig. 7b) as well as a sentence referring to it in the text.

**Referee:** Line 371: "They document that…" Who are **they**?
**Reply:** The words "*They document*" have been replaced by "*This implies*".

**Referee:** Line 405-413. With the regression analysis for u=a*dH+b, the values a will determine the strength of the variability while b determines the bias. Thus, it is the **variability** of the surface velocity that is underestimated.
**Reply:** We have added a sentence: "*This might be due to a large bias, b in Eq. (3), for this site but inspection of individual daily velocity profiles does not support that (Fig. 9 in Hansen et al., 2018).*"

**Referee:** Line 439-440: According to Figure 6, zero velocity are not reached at several ADCP-locations. Is the vertical integration done to bottom when the velocity does not reach zero?
**Reply:** The sentence before Eq. (7) has been changed to: "*Sites IA, IB, and IE have inflow throughout the water column, on average (Fig. 7a), and the integration is down to the bottom. For site IW, we only integrate down to the depth, $z = z_0$, where the average cross-ridge velocity becomes zero:*"

**Referee:** Table 7. The mean values of $D_{eq}$ and $L_{eq}$ are presented but I assume that there might be large temporal variation. If standard deviations or errors can be included, this will give some indications on the sensitivity of the method.
**Reply:** For ADCP site IW, we now write: "*$L_{Eq} = (12 \pm 4)$ km, where the uncertainty is determined by the uncertainty of $\alpha_{Reg}$*". For the other sites on the IFR, we write: "*For these sites, the relative uncertainty of $L_{Eq}$ is higher (between 38 % and 75 %), and $L_{Eq}$ has been set equal to the interval width, L.*". For $D_{Eq}$, we could not find any objective way to derive any uncertainty estimate. We have, however, added a sentence to Sect. 7.4 (old line number 832): "*Also, the many uncertainties involved make the numbers in the bottom row of Table 4 rough estimates.*"

**Referee:** Figure 11. Is the altimetric offset calculated from the averaged surface velocities from ADCPs?
**Reply:** The caption for Fig. 11 (new Fig. 12) has been clarified by the following text: **"***Optimized values for the Altimetric offset, $U_k^0$, in each altimetry interval are shown by the thick continuous red line. The value for $U_2^0$ is based on ADCP NI (Table B2). $U_3^0$ and $U_4^0$ are based on linear combinations of surface velocities from two or three ADCPs (Sect. 3). $U_5^0$ and $U_6^0$ are combined estimates from NC and NH (Table B2) and the geostrophic method. $U_7^0$ is based on the geostrophic method.*"*

**Referee:** Line 532: Please include a reference for the depth of AW = 4 $^\circ$C.
**Reply:** We now refer to the new Sect. 1.3.

**Referee:** Line 573: see my comment on Line 532.
**Reply:** We now refer to the new Sect. 1.3.

**Referee:** Line 772: "… surface AND deeper…"
**Reply:** Has been corrected

**Referee:** Line 776: See my major comment.
**Reply:** Hopefully, the new Sect. 1.3 with information on Atlantic water clarifies here.
**Appendix**

**Referee:** Eq. A1: why 500 m? Why not use depth of the Atlantic layer or bottom depth?
**Reply:** Our reason for doing this is that the deep boundary of the Atlantic layer is in general sloping. This was not well explained, however, so now the following text has been inserted before Eq. (A1): "*Within each altimetry interval, k (spanning $A_k$–$A_{k+1}$), $U_k(z,t)$ is the eastward velocity at depth z and time t, horizontally averaged within the interval. The contribution to Q(t) from this interval is found by integrating (summing) the velocity down to the deep boundary of the Atlantic layer (bottom or 4 °C isotherm) and multiplying by the interval width. The deep boundary is, however, in general not horizontal. To account for this, we introduce a parameter $W_k(z,t)$, which is the width of Atlantic water within altimetry interval k at depth z and time t. With this definition, the volume transport is:*". The paragraph after Eq. (A1) has been modified accordingly.

---

## Author Response (AR2)

**Responses to editor comments**

We thank the editor for positive and constructive comments.

**Editor:** The paper is very long, and as noted by both reviewers, it tends to intermingle methods, results and discussion. It would be good if you could read it through again yourselves critically, to see if you can identify any further shifting of material that would make the paper flow more smoothly. I suspect this will help it to be well-cited.

**Reply:** We are aware that this manuscript is long and will require considerable effort from the reader to gain full benefit. It has two main topics: 1) the ability of satellite altimetry to represent surface velocity and other oceanographic features and 2) the flow patterns, variability, and transport of the IF-inflow. At an early stage, we considered splitting the results into two separate manuscripts, each focusing on one of these topics. We concluded, however, that the two topics are so strongly interlinked that this would not improve the overall product. Before the original submission, we have also considered other versions with different ordering of the content, but ran into difficulties with having to cite results that were presented later in the manuscript. Your comments as well as those of the two referees have clearly helped to link the various parts of the manuscript together and improve the flow, but we do recognize that the text is still long and complicated. We do not see any way to avoid this without losing essential parts. We have, however, added the following paragraph at the end of Sect. 1: "*Several different topics are addressed in this manuscript, although they are interlinked.* **Readers who do not want all the details may benefit from starting in Sect. 7 and referring to the earlier sections as needed".** (New lines: 143-144).

**Editor:** L108-124 This is useful material added, but it feels to me like it is in the wrong place. This is not Introduction – rather it is justification for your choice of methods, so at least some of this should be moved into section 2. Please review which pieces of this material are really introduction, and which are methods.

**Reply:** The original Sect. 1.3 has been abbreviated by removing the text from original line 104 to original line 124. This section now ends with the added sentence: "*Hansen et al. (2015) therefore decided instead to use the 4 °C isotherm and the 35.0 isohaline to define the boundaries of Atlantic water extent on the N-section*". (New lines: 106-107).

The deleted text in Sect. 1.3 has been moved to a new section (new lines: 248-271) within methods. The new section has the heading: "*2.6 Determination of Atlantic water extent on the monitoring section*" and is introduced by a new paragraph: *"On the N-section, used for transport monitoring, water of Arctic origin is found adjacent to and mixed with the Atlantic water. To enable calculation of Atlantic water transport through the section, this study uses (temporally varying)* **Atlantic water boundaries,** *within which all of the water is assumed to be of Atlantic origin, with no Atlantic water outside of the boundaries*". (New lines: 249-252).

**Editor:** L186. For clarity, add a sentence to say something like "Therefore we use the travel time to deduce the isotherm depth, with results shown in section X.Y".

**Reply:** We have added the sentence: "*Estimates of travel time from the two PIES deployments will therefore be used to calculate monthly averaged isotherm depth (Sect. 6.3 and Sect. 7.3)*". (New lines: 171-172).

**Editor:** L329 It would be helpful if you could add a sentence (or two) here summarising the results of section 3 for readers, and/or saying that the implications will be discussed in sections 6 or 7. At the moment each section seems to stop abruptly and the logical connections between sections are not always clear. Signposting would help your readers.

**Reply:**
The paragraph at the end of Sect. 3 (original lines 327-329) has been rewritten and now ends with the text: "*This result is further discussed in Sect. 7.1. The observational verification of geostrophic balance on monthly timescales when using the new SLA data is also a basic precondition for other results in this manuscript, such as the flow across the IFR (Sect. 5 and Sect. 7.2) and the calculation of transport (Sect. 6 and Sect. 7.3–7.6)*". (New lines: 340-343).

**Editor:** L17 and L934 You use the phrase conversion factor here, but you never refer to the phrase conversion factor in the text. It will not be clear to readers who just read the abstract or the conclusions what you mean by this, so this should be clarified in both places. One might expect using the standard geostrophic equation f V = g tan(i) that the slope of the sea surface is directly related to the surface geostrophic flow without any need for a "conversion factor", so this needs some clarification.

**Reply:** On the original lines 17 and 934, the term "*conversion factor*" has been replaced by: "*conversion factor between sea level slope and surface velocity*". (New lines: 17-18 and 950).

In addition, the paragraph at the end of Sect. 3 (original lines 327-329) has been rewritten and now includes a definition of this term: "*The regression coefficient, $\alpha_{Reg}$, in Table 2 is the observationally determined conversion factor between anomalies of sea level slope and surface velocity, Eq. (3). In the geostrophic approximation, this conversion factor should have the value given by $\alpha_{Th} \equiv g/(f \cdot L)$. Table 2 demonstrates that this is the case when using the new SLA data to calculate sea level slope, but not when the old SLA data are used*". (New lines: 337-340).

The term has also been introduced to the discussion in Sect. 7.1 where the sentence: "*A high correlation between two time series means that they are linearly related, but the coefficients may not necessarily be according to theory*" has been replaced by: "*A high correlation between ADCP surface velocity and SLA-difference means that they are linearly related, but this does not guarantee that the conversion factor between sea level slope and surface velocity is according to theory*". (New lines: 672-673).

**Editor:** L941 Can you be more quantitative than "slight"? e.g. giving a number for Sv/decade, or % increase? Likewise for the heat transport?

**Reply:** The sentence: "*Over the 29 years of monitoring, the IF-inflow had a slight increase in volume transport and also an increase in heat transport relative to a temperature of 0 °C*" has been replaced by: "*Over the 29 years of monitoring, the IF-inflow had a slight (9 %) increase in volume transport and also an increase (13 %) in heat transport relative to a temperature of 0 °C*". (New line: 957).

**Editor:** L942 Please replace "an hypothesis" by "the hypothesis proposed by Olson et al. (2016)" so that the conclusions section stands alone.

**Reply:** The text: "*an hypothesis, previously suggested to explain*" has been replaced by: "*the hypothesis proposed by Olsen et al. (2016) to explain*". (New lines: 958-959).